# The βI domain promotes active β1 integrin clustering into mature adhesion sites

Giulia Mana[1,2,*] , Donatella Valdembri[1,2,*] , Janet A Askari[3], Zhenhai Li[4] , Patrick Caswell[3] , Cheng Zhu[4] , Martin J Humphries[3] , Christoph Ballestrem[3] , Guido Serini[1,2]

**Modulation of integrin function is required in many physiological and pathological settings, such as angiogenesis and cancer. Integrin allosteric changes, clustering, and trafficking cooperate to regulate cell adhesion and motility on extracellular matrix proteins via mechanisms that are partly defined. By exploiting four monoclonal antibodies recognizing distinct conformational epitopes, we show that in endothelial cells (ECs), the extracellular βI domain, but not the hybrid or I-EGF2 domain of active β1 integrins, promotes their FAK-regulated clustering into tensin 1–containing fibrillar adhesions and impairs their endocytosis. In this regard, the βI domain–dependent clustering of active β1 integrins is necessary to favor fibronectin-elicited directional EC motility, which cannot be effectively promoted by β1 integrin conformational activation alone.**

## Introduction

Integrins are major extracellular matrix (ECM) surface receptors that, by modulating cytoskeletal flux and associated signaling pathways, control cell motility, survival, proliferation, and differentiation during embryonic development (Wickström et al, 2011), immune reactions, hemostasis, angiogenesis, and cancer (Weis & Cheresh, 2011; Hamidi & Ivaska, 2018; Cooper & Giancotti, 2019). The dynamic regulation of the physical interactions of integrins with the ECM is necessary for physiological function, and it is disrupted in pathological settings (Seguin et al, 2015). Such a plastic behavior depends on the ability of integrins to undergo conformational changes, clustering, and endocytosis (Moreno-Layseca et al, 2019; Sun et al, 2019). Although much is known about how integrin conformation and traffic are modulated (Moreno-Layseca et al, 2019; Sun et al, 2019; Chastney et al, 2021), less is known about the molecular mechanisms that control integrin clustering.

Integrins exist in at least two major structural classes: an extended-open conformation endowed with high affinity for ECM ligands and a bent-closed, low-affinity conformation (Luo & Springer, 2006; Campbell & Humphries, 2011). In the fully active extended-open conformation, the extracellular headpiece of the integrin β subunit moves away from the leg (integrin extension), its βI domain rearranges to bind the ECM ligand with high affinity (headpiece opening), and two NPxY motifs in the cytoplasmic domain directly interact with the phosphotyrosine-binding (PTB) domains of the cytoskeleton-associated adaptors talin, kindlin, and tensin (Kim et al, 2011; Calderwood et al, 2013; Rognoni et al, 2016; Georgiadou et al, 2017; Sun et al, 2019; Atherton et al, 2022). The ability to simultaneously interact with multivalent ECM ligands (Schwarzbauer & DeSimone, 2011) and cytoskeletal adaptors (Calderwood et al, 2013; Sun et al, 2019; Bu et al, 2021) drives active integrin clustering and retention within adhesion sites. The extent of integrin clustering, usually defined as avidity (Carman & Springer, 2003; Geiger et al, 2009), and the length scale of inter-integrin spacing (Arnold et al, 2004; Cavalcanti-Adam et al, 2007; Geiger et al, 2009; Young et al, 2016) have been employed as parameters to quantify the formation of adhesive contacts.

Integrin endo-exocytic trafficking also regulates cell-to-ECM adhesion dynamics (Paul et al, 2015; Moreno-Layseca et al, 2019). Ligand-bound (active) integrin endocytosis requires the cleavage of ECM polymers by matrix metalloproteinases (Shi & Sottile, 2011), and the association of NPxY motifs of integrin β subunit cytotails to PTB domain-containing endocytic adaptors (De Franceschi et al, 2015; Mana et al, 2020; Samarelli et al, 2020). Indeed, ligand-bound/active integrin internalization is fostered by shifting their association from cytoskeletal to endocytic adaptors, such as Dab2 (Yu et al, 2015; Cao et al, 2020). Because polymerization of multivalent ECM ligands impairs integrin endocytosis (Shi & Sottile, 2011; Stehbens & Wittmann, 2012; Mana et al, 2020; Elkhatib et al, 2021), clustering may play a major role in the control of active integrin internalization and function.

The importance of integrin clustering in the formation of adhesion complexes is well established (Geiger et al, 2009; Young et al, 2016). In the case of β3 integrins, mature large focal adhesions arise from the aggregation of individual smaller nascent adhesions that occurs via yet unknown mechanisms (Changede et al, 2015; Changede & Sheetz, 2017). Because cytoskeletal adaptors, such as

---

[1]Candiolo Cancer Institute - FPO, IRCCS, Candiolo (TO), Italy [2]Department of Oncology, University of Torino School of Medicine, Candiolo (TO), Italy [3]Wellcome Centre for Cell-Matrix Research, Faculty of Biology, Medicine and Health, University of Manchester, Manchester, UK [4]Wallace H. Coulter Department of Biomedical Engineering, Georgia Institute of Technology and Emory University, Atlanta, GA, USA

Correspondence: guido.serini@ircc.it
*Giulia Mana and Donatella Valdembri contributed equally to this work

talin (Saltel et al, 2009; Kukkurainen et al, 2020) and kindlin (Ye et al, 2013; Theodosiou et al, 2016), promote both integrin conformational activation and clustering, it was difficult to experimentally dissect the role of these two mechanisms in the assembly of adhesion sites. In addition, it is unknown if and how the activation state of the extracellular domains of integrin subunits may play a role in the control of clustering. Here, by treating live endothelial cells (ECs) with mAbs directed against conformational epitopes located in distinct, functionally relevant regions of the active conformer of the integrin $\beta$1 subunit, we provide evidence that the N-terminal $\beta$I domain, but not the hybrid or the I-EGF2 domain, promotes the FAK-regulated clustering of active $\beta$1 integrins into mature adhesion sites and the haptotactic migration of ECs towards the ECM.

## Results

### Extracellular $\beta$I domain promotes active $\beta$1 integrin clustering in living ECs

To pinpoint the different roles that integrin conformation and clustering play in the assembly of adhesion sites, we sought to exploit the steric and functional hindrance that Abs recognizing extracellular epitopes may display towards adhesion receptors. We selected a range of mAbs that stabilize the active conformation of $\beta$1 integrins and promote cell spreading on the $\alpha$5$\beta$1 integrin–specific ligand fibronectin (FN). To stabilize the open headpiece, we selected 12G10 (Takada & Puzon, 1993; Mould et al, 1998, 2002) and HUTS4 (Luque et al, 1996), whose epitopes lie in the $\beta$I domain and hybrid domain, respectively (Fig 1A). Conformational modifications of sites close to these two epitopes are involved in both headpiece opening ($\beta$I domain) and acquisition of the extended-open conformation (hybrid domain) during $\beta$1 integrin conformational activation (Stephanie et al, 2021). During integrin headpiece opening, the $\alpha$1/$\alpha$1'-helix of the $\beta$I domain straightens and undergoes an inward movement that is coupled to a downward piston-like shift of the $\alpha$7-helix. In the extended-open conformation, $\alpha$7-helix movement causes the hybrid domain to swing out and alter its angle with the $\beta$I domain $\alpha$1'-helix from obtuse to straight (Stephanie et al, 2021). To stabilize lower leg extension, we selected 9EG7 (Bazzoni et al, 1995; Askari et al, 2010), the epitope of which is located in the I-EGF2 domain of the active $\beta$1 integrin subunit (Fig 1A).

We first determined the subcellular localization of active $\beta$1 integrin subunit epitopes, as recognized by these three mAbs, in fixed primary human ECs isolated from umbilical cord veins. Fluorescence confocal microscopy showed that all three mAbs labeled active $\beta$1 integrins in fibrillar adhesions (Zamir & Geiger, 2001) (Fig 1B). Next, we assessed the subcellular distribution of the mAb epitopes after a 10-min incubation on live ECs, followed by fixation and analysis by confocal fluorescence microscopy. Although 9EG7- and HUTS4-bound active $\beta$1 integrins were found in fibrillar adhesions, 12G10-bound active $\beta$1 integrins were located within and outside significantly shorter ECM adhesions (Fig 1C), which we speculated were derived from the fragmentation of pre-existing fibrillar adhesions. To test this possibility, we analyzed the localization of fluorescently labeled 9EG7-Alexa Fluor 488 as a

function of time (1.5–10 min) on ECs cultured in the absence (Fig 1D, *upper row* and Video 1) or the presence of 12G10-Alexa Fluor 647 (Fig 1D, *lower rows* and Video 2). Pre-incubation of live ECs with 12G10-Alexa 647 impaired the localization of 9EG7-Alexa 488 to fibrillar adhesions (Fig 1D, *lower rows* and Video 2). Next, we quantified the impact of 12G10 on the clustering and spatial organization of 9EG7-labeled integrins in fibrillar adhesions of living ECs by both standard (Fig S1) and super-resolution time-gated stimulated emission depletion ($g$-STED) confocal microscopy (Fig 1E, *left panels*). Using computer-assisted automated analyses, we morphologically categorized clusters of 9EG7$^+$ active $\beta$1 integrin according to their shape factor (SF) as elongated (SF < 0.5) or round (SF ≥ 0.5) (Fig 1E, *right panels*). We found that incubating cultured ECs with 12G10 shortened the maximum Feret's diameter (mFD), that is, the longest distance between any two points along a single cluster of elongated 9EG7$^+$ active $\beta$1 integrin clusters (Fig 1E). Analogously, live incubation of ECs with mAb TS2/16 (Takada & Puzon, 1993), which recognizes an epitope overlapping to that targeted by 12G10 in the $\beta$I domain of active $\beta$1 integrin subunit (Mould et al, 1995) and stabilizes the open headpiece (Su et al, 2016), also reduced the mFD of elongated 9EG7$^+$ active $\beta$1 integrin clusters (Fig 1E). In fibrillar adhesions of both ECs (Mana et al, 2016) and fibroblasts (Pankov et al, 2000; Clark et al, 2005), the $\beta$1 integrin subunit exists as an $\alpha$5$\beta$1 heterodimer that is a major FN receptor. Therefore, we evaluated the impact of 12G10 on the subcellular patterning of extended active $\alpha$5 integrin subunit, as recognized by the SNAKA51 mAb, whose epitope lies in the lower leg calf domains of active $\alpha$5 integrin (Clark et al, 2005). As observed for 9EG7$^+$ active $\beta$1 integrins (Fig 1E), we found that incubating cultured ECs with 12G10 also shortened the mFD of elongated SNAKA51$^+$ active $\alpha$5 integrin clusters (Fig S2A). On the contrary, 12G10 did not influence the number, mean area, or mFD of $\beta$3 integrin$^+$ adhesion sites of living ECs (Fig S2B). Taken together, these findings suggest that the $\beta$I domain, but not the hybrid or lower leg domain, is involved in the aggregation of the active conformer of the FN receptor $\alpha$5$\beta$1 integrin and its accumulation into fibrillar adhesions.

Because clustered active $\alpha$5$\beta$1 integrins are tethering sites that promote the polymerization of soluble FN dimers into an insoluble fibrillar network (Schwarzbauer & DeSimone, 2011), next we assessed the influence of 12G10 on the incorporation of soluble FN into polymeric FN fibrils. To achieve this aim, rhodamine-labeled soluble FN was added to cultured ECs for 1 h and then followed or not by a 10-min incubation with 12G10, fixation, and analysis by confocal fluorescence microscopy. We found that 12G10 clearly disrupted the incorporation of soluble rhodamine-FN into fibrils while promoting its accumulation into punctate structures (Fig S2C) that were bona fide endosomes (see below).

The observed disaggregation of 9EG7$^+$ and SNAKA51$^+$ active $\alpha$5$\beta$1 integrin–containing fibrillar adhesions elicited by 12G10 might be caused by steric hindrance because of the simultaneous incubation of ECs with two mAbs. To test this possibility, we determined the effect of treating live cells with 9EG7 or 12G10 alone on fibrillar adhesions by $g$-STED confocal microscopy with tensin 1 as a fibrillar adhesion marker (Pankov et al, 2000). We found that both in control and 9EG7-treated cells, tensin 1 localized in fibrillar adhesions and displayed similar mFDs, whereas 12G10 treatment reduced the mFD of tensin 1 (Fig 2A). Thus, 12G10 treatment alone is sufficient to

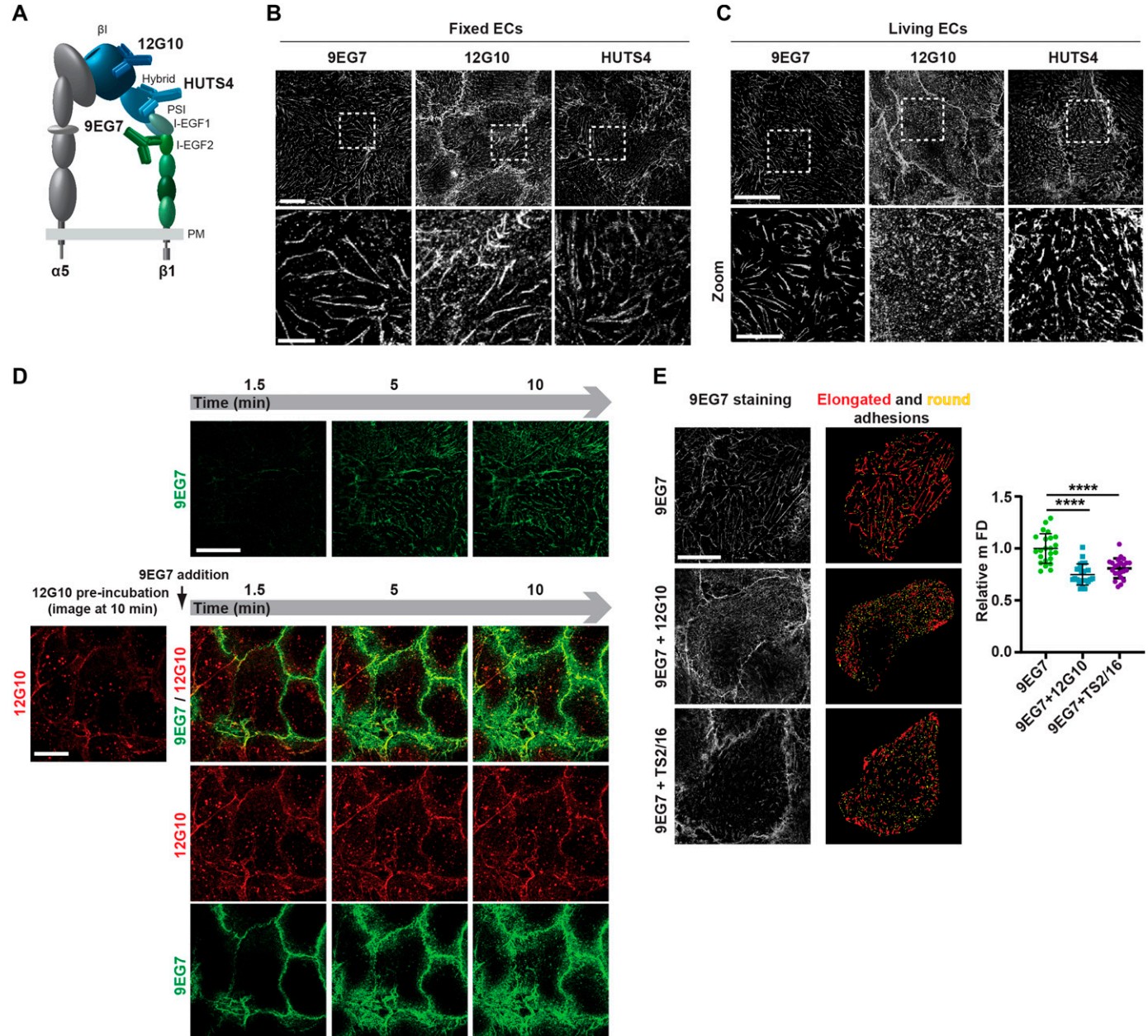

**Figure 1. Anti-βI domain mAb 12G10 hampers mAb 9EG7⁺ active β1 integrin clustering in living ECs.**

**(A)** Domain architecture of an active (open headpiece/extended) integrin α5β1 heterodimer. α5 Subunit is grey; β1 subunit headpiece and leg are, respectively, in shades of blue and green. The localization of three different mAb epitopes, exposed only in the conformationally active β1 subunit, is represented. Epitopes of mAb 12G10 and mAb HUTS4, respectively, lie in the βI domain and hybrid domain of the headpiece, whereas mAb 9EG7 epitope is in the I-EGF2 domain. **(B)** Confocal immunofluorescence microscopy analysis of the subcellular localization of the three different anti-active β1 integrin mAbs employed to stain fixed ECs. All three mAbs bind to active β1 integrins mainly located within typical elongated fibrillar adhesions. Scale bar 20 μm; magnification scale bar 10 μm. **(C)** Confocal immunofluorescence microscopy analysis of anti-active β1 integrin mAbs localization after 10 min of incubation on living ECs. Anti-I-EGF2 domain mAb 9EG7 preferentially binds to active β1 integrins located within elongated fibrillar adhesions, whereas anti-βI domain mAb 12G10 recognizes active β1 integrins located both outside and inside highly fragmented and tiny adhesions. Similar to mAb 9EG7, the anti-hybrid domain mAb HUTS4 preferentially binds to active β1 integrins located within elongated fibrillar adhesions, hinting that anti-active headpiece mAb-elicited fragmentation specifically depends on βI domain binding. Scale bar 20 μm; magnification scale bar 10 μm. **(D)** Selected frames from Video 1 (top row) and Video 2 (bottom rows), respectively, illustrating dynamic mAb 9EG7-Alexa Fluor 488 binding to active β1 integrins over time upon live incubation on ECs either in the absence (top row) or in the presence (bottom rows) of mAb 12G10–Alexa Fluor 647. When incubated alone (top row), mAb 9EG7–Alexa Fluor 488 preferentially binds active β1 integrins located within fibrillar adhesions and remains stable over time. When mAb 12G10–Alexa Fluor 647 is pre-incubated on ECs, mAb 9EG7–Alexa Fluor 488 does no longer localize in fibrillar adhesions. Scale bar 20 μm. **(E)** Representative g-STED confocal microscopy pictures of anti-active β1 integrin 9EG7 mAb localization after 10-min incubation on living ECs either in the absence (top left panel) or the presence (middle left panel) of 12G10 or TS2/16 (bottom left panel). To thoroughly analyze the morphology of ECM adhesion sites, g-STED confocal images were acquired close to the basal EC surface. 9EG7-labeled adhesions were then analyzed with ImageJ software (right panels) and classified, according to their shape factor (SF), into elongated (red) and round (yellow) structures. 9EG7-labeled adhesions were classified as elongated, if their SF was < 0.5, and round, if the SF was ≥ 0.5. Scale bar 20 μm. The maximum Feret's diameter was

disrupt tensin 1–containing fibrillar adhesions in living ECs. The biological effects of some anti-β1 integrin Abs were previously ascribed to the interaction of their constant fragment (Fc) moieties with Fc receptors expressed on the surface of target cells (Tsuchida et al, 1997). To test whether the 12G10-elicited disaggregation of fibrillar adhesions in living ECs involves Fc receptor signaling and/or Ab bivalency, we compared the effects of the whole mAb with those of equivalent amounts of its Fab fragments (Fig 2A). We found that, analogously to the whole mAb, 12G10-Fab fragments reduced the mFD of elongated tensin 1+ clusters (Fig 2A). Hence, the disrupting effects of 12G10 on fibrillar adhesions do not rely on Fc-dependent signaling or Ab bivalency. Furthermore, similar to what was observed with 12G10 mAb and Fab, live incubation of ECs with TS2/16 mAb also reduced mFD of elongated tensin 1+ clusters (Fig 2A).

The binding between an integrin and its ECM ligand behaves as a catch bond, that is, a non-covalent link whose lifetime increases with the increasing tensile force applied to it (Friedland et al, 2009; Kong et al, 2009; Chen et al, 2016). 12G10 considerably shifts α5β1 integrin-FN catch bond to a lower force range (Kong et al, 2009); however, it is not known if 9EG7 has a similar activity. Therefore, we evaluated whether fibrillar adhesion fragmentation caused by live EC incubation with 12G10, but not 9EG7, depends on the increased lifetime of the α5β1 integrin/FN bond at low forces (Kong et al, 2009). By means of atomic force microscopy (AFM) experiments, we observed that, similar to 12G10, 9EG7 significantly lessened the force range of α5β1 integrin-FN catch bonds (Fig 2B). Hence, the fragmenting activity of 12G10 cannot be ascribed to differences in its ability to prolong the lifetime of α5β1 integrin/FN catch bond at low forces.

### Stimulating active β1 integrin clustering counteracts the dismantling effect of βI domain interference on fibrillar adhesions

Our data supported the concept that live incubation with the anti-βI domain mAb 12G10 may disrupt endothelial fibrillar adhesions by interfering with active β1 integrin clustering. To directly address this issue, we evaluated the effect of stabilizing the aggregation of active β1 integrins in fibrillar adhesions. Tensin 1 is a cytoskeletal protein that links (α5)β1 integrins to the actin cytoskeleton, and increasing tensin 1 expression promotes FN fibril formation (Georgiadou et al, 2017). We therefore promoted (α5)β1 integrin clustering by overexpressing EGFP-tagged tensin 1 and tested its ability to oppose 12G10-Alexa Fluor 647–triggered disaggregation of active β1 integrin–containing fibrillar adhesions in living ECs. Time-lapse fluorescence confocal microscopy revealed that, differently from control ECs treated with 12G10-Alexa Fluor 647 (Fig 3A and Video 3), in EGFP-tensin 1–transfected ECs, 12G10-Alexa Fluor 647 bound active β1 integrin in tensin 1+ fibrillar adhesions without eliciting their dismantling (Fig 3B and Video 4). Similarly, we found that pre-incubating live ECs with 9EG7-Alexa Fluor 488 stabilized the clustering of active β1 integrins in fibrillar adhesions such that they

were no longer disrupted by the subsequent addition of 12G10-Alexa Fluor 647 (Fig 3C and Video 5). Next, we verified whether the counteracting effect of 9EG7 could be simply because of its ability to stabilize the active conformation of β1 integrins or whether the dimeric nature of 9EG7 mAb was instead required to strengthen the clustering of active β1 integrins. To achieve this aim, we compared the effect of the dimeric intact 9EG7 mAb with that of its monomeric Fab. We found that pre-incubating cultured ECs with the dimeric intact form, but not with the monomeric Fab of 9EG7-Alexa Fluor 488, counteracted the disassembly of active β1 integrin–containing fibrillar adhesions by 12G10-Alexa Fluor 647 (Fig S3). Taken together, these data demonstrate that the fibrillar adhesion–disrupting activity of 12G10 can be prevented by increasing integrin activity from either the outside or inside of the cell.

FAK is a key regulator of the turnover of focal adhesions (Tomar & Schlaepfer, 2009). Autophosphorylation on Tyr 397 increases the residency of FAK at focal adhesions, eventually inducing their disassembly through different mechanisms (Franco et al, 2004; Webb et al, 2004; Ezratty et al, 2005; Hamadi et al, 2005). Initially, we tested the role of the tyrosine kinase activity of FAK in the turnover of fibrillar adhesions. We transfected tensin 1-EGFP into ECs and quantified its turnover at fibrillar adhesions by FRAP after treatment with DMSO, as a control, or with the FAK inhibitor PF-562271 (Roberts et al, 2008). FRAP analysis showed that the tensin 1-EGFP mobile fraction was reduced in ECs upon FAK inhibition by PF-562271 (Fig 3D), indicating that FAK not only stimulates the disassembly of focal adhesions (Franco et al, 2004; Webb et al, 2004; Ezratty et al, 2005; Hamadi et al, 2005) but also fibrillar adhesions. Next, we tested if impairing fibrillar adhesion turnover by inhibiting the enzymatic activity of FAK could counteract the disaggregating activity of 12G10. ECs were first pre-treated with either DMSO (as control) or PF-562271 and then incubated with 12G10. Confocal microscopy analysis demonstrated that PF-562271 reduced the 12G10-elicited shortening of the mFD of elongated active β1 integrin clusters (Fig 3E). However, whereas FAK tyrosine kinase activity was necessary for the disassembly of fibrillar adhesions by 12G10, Western blot analyses revealed that neither 12G10 nor 9EG7 modulated FAK phosphorylation (Fig 3F). Therefore, the disassembly of fibrillar adhesions caused by 12G10 does not require any increase, but only the permissive steady activity of FAK, which relocates from ECM adhesions to intercellular contacts of live ECs treated with 12G10 (Fig 3E).

Taken together, these findings suggest that 12G10 disrupts endothelial fibrillar adhesions by hampering the βI domain- and tensin 1-mediated clustering of conformationally active β1 integrins in an FAK-dependent manner.

### βI domain interference supports active β1 integrin endocytosis from ECM adhesions and impairs ECM-elicited EC migration

Because stimulating active β1 integrin clustering by tensin 1 overexpression counteracted the 12G10-induced disassembly of

measured to quantify the morphological features of 9EG7+ elongated structures. Compared with control ECs incubated live with 9EG7 alone, 9EG7+ elongated structures were significantly shortened in the presence of 12G10 or TS2/16. Data are mean ± SD, n ≥ 20 cells per condition pooled from two independent experiments. Statistical analysis: unpaired t test, P ≤ 0.0001 ****.
Source data are available for this figure.

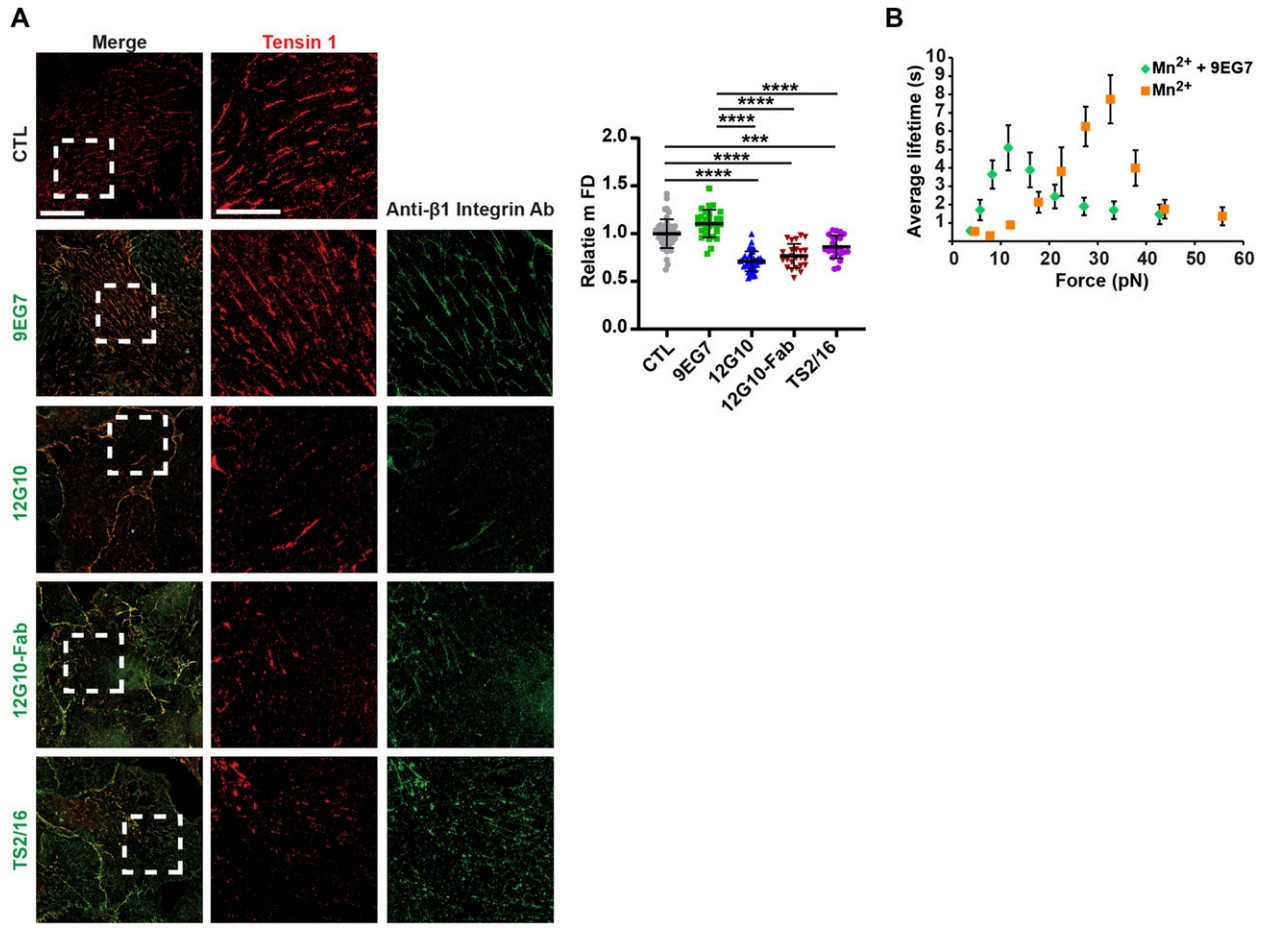

**Figure 2. Anti-βI domain mAb 12G10 hampers tensin 1⁺ active β1 integrin clustering in living ECs.**
**(A)** Representative *g*-STED confocal microscopy analysis of tensin 1 localization in ECs that were incubated or not for 10 min with the anti-active β1 integrin mAb 9EG7 or mAb 12G10 or the Fab fragment of mAb 12G10 (12G10-Fab) or mAb TS2/16. Scale bar 20 μm; magnification scale bar 10 μm. When compared with untreated control (CTL) ECs or those treated with mAb 9EG7, the mFD of tensin 1⁺ fibrillar adhesions was significantly reduced in ECs treated with either mAb 12G10, 12G10-Fab or mAb TS2/16. Data are mean ± SD, n ≥ 22 cells per condition pooled from three independent experiments. The number of structures was normalized on cell area and on those in control cells. Statistical analysis: one-way ANOVA and Bonferroni's post hoc analysis; *P* ≤ 0.0001 ****. **(B)** MAb 9EG7 affects the lifetime of FN–α5β1 bonds. AFM measurement of mAb 9EG7 effect on force-dependent lifetime of single bonds between a FNIII$_{7-10}$ fragment and an integrin α5β1-Fc fusion protein. Lifetime versus force plots of α5β1-Fc–functionalized Petri dish dissociating from FNIII$_{7-10}$-coated cantilever tips in Mn$^{2+}$ either in the absence (grey) or the presence (green) of 10 μg/ml mAb 9EG7 mAb. Data are mean ± SEM of several tens to several hundreds of measurements per point.
Source data are available for this figure.

fibrillar adhesions, 12G10 may function by displacing cytoskeletal adaptors, such as tensin 1, in favor of NPXY-binding endocytic adaptors (De Franceschi et al, 2015; Yu et al, 2015; Cao et al, 2020; Mana et al, 2020; Samarelli et al, 2020), we tested the effects of 12G10-stimulated disaggregation of fibrillar adhesions on endocytosis of active β1 integrins, as recognized by mAb 9EG7. We incubated ECs with 9EG7 alone or in combination with 12G10 for 10 min and, to eliminate non-endocytosed mAb from the cell surface, ECs were acid washed and fixed. Subsequently, the degree of colocalization between endocytosed 9EG7⁺ active β1 integrins and the early endosome marker early endosome antigen 1 (EEA1) was quantified by confocal fluorescence microscopy and Pearson's correlation coefficient analysis. In agreement with a previous report (Arjonen et al, 2012), 9EG7-bound integrins were internalized from the EC surface into EEA1⁺ early endosomes (Fig 4A); however, this was increased by the simultaneous incubation with 12G10 (Fig 4A).

These findings were validated using biochemical assays in which surface biotinylated and internalized active β1 integrins were quantified by capture ELISA assays using 9EG7 (Sandri et al, 2012). 12G10 again caused an increase in the endocytosis of endogenous 9EG7⁺ active β1 integrins from the surface of ECs (Fig 4B). As observed for 9EG7⁺ active β1 integrins (Fig 4A), we found that incubating cultured ECs with 12G10 also increased the internalization of SNAKA51⁺ active α5 integrin from the cell surface into EEA1⁺ early endosomes (Fig S4).

Finally, we exploited the different functional properties of the anti-βI domain mAb 12G10 and the anti-I-EGF2 domain mAb 9EG7 to test the role of conformationally active β1 integrin clustering in the control of ECM-elicited EC directional migration. Impedance-based time-lapse haptotaxis assays showed that, when compared with control, 9EG7, but not 12G10, promoted the directional motility of ECs towards FN (Fig 4C). Furthermore, when we incubated migrating ECs with 9EG7 in combination with 12G10, the simultaneous

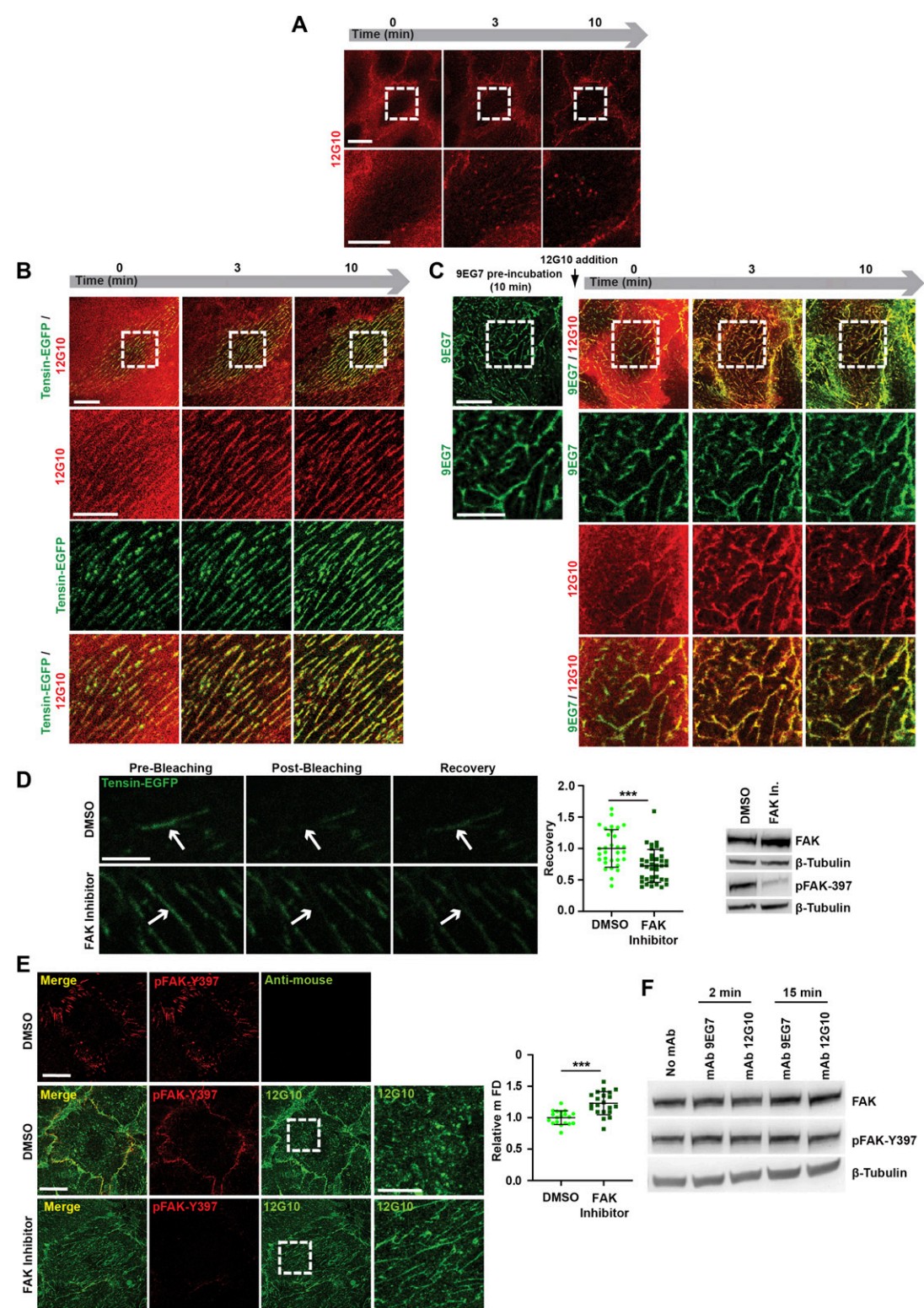

**Figure 3. Strategies aimed at stabilizing the clustering of active β1 integrins neutralize the anti-βI domain mAb 12G10 disassembling effect on fibrillar adhesions of living ECs.**
**(A, B, C)** Selected frames from Video 3 (A), Video 4 (B), and Video 5 (C), monitoring mAb 12G10 (*red*) binding to active β1 integrins on living: control ECs (A); ECs previously oligofected with tensin-EGFP (B); ECs pre-incubated for 10 min with mAb 9EG7 (C). **(A, B, C)** As expected, mAb 12G10 mAb binds active β1 integrins located within and outside fragmented adhesion sites (A), but, either when Tensin-EGFP is overexpressed (B) or upon pre-incubation with mAb 9EG7 (C), mAb 12G10 mAb localizes instead within fibrillar adhesions that remain stable over time. Scale bar 20 μm; magnification scale bar 10 μm. **(D)** Representative confocal images showing pre-bleaching, post-bleaching, and recovery on the region of interest (indicated by arrows) of tensin-EGFP–positive fibrillar adhesions in ECs treated with DMSO (as control) or with the FAK inhibitor PF-562271. Scale bar 5 μm. Recovery rate was measured, and data were normalized by employing as reference the fluorescence intensity acquired on the same ROI

presence of 12G10 dampened the stimulatory effect of 9EG7 (Fig 4D). Thus, directional EC migration towards the ECM relies on both β1 integrin conformational activation and clustering.

# Discussion

The functional regulation of integrin receptors depends on their allosteric activation, cell surface clustering, and endo-exocytic trafficking (Moreno-Layseca et al, 2019; Sun et al, 2019; Chastney et al, 2021), but it is unclear how these three functional processes relate to each other (Iwamoto & Calderwood, 2015). By exploiting three distinct mAbs that, although equivalently effective in stabilizing active conformers of β1 integrins (Byron et al, 2009), differentially hinder integrin clustering, we were able to disentangle the roles of integrin conformation and clustering. Our major finding is that (α5)β1 integrin activation alone is not sufficient to drive fibrillar adhesion formation, incorporation of soluble FN dimers into fibrillar polymers, and increase ECM-dependent EC migration. Indeed, all three functions require the aggregation of conformationally active β1 integrins at the plasma membrane. Furthermore, nonclustered active (α5)β1 integrins are rapidly endocytosed, further impairing their incorporation into fibrillar adhesions.

Integrin conformational activation is characterized by the separation of α and β subunit cytoplasmic tails that is stabilized by the binding of PTB domain–containing actin cytoskeleton adaptors, such as talin (García-Alvarez et al, 2003) or tensin (McCleverty et al, 2007), to the membrane proximal NPxY motif of integrin β subunits. Both talin (Goldmann et al, 1994) and tensin (Lo et al, 1994) exist as homodimers, and microscopy-based stoichiometric analysis of living cells indicated that clustering of integrins that gives rise to short-lived nascent adhesions originates with the interaction of two active integrin-kindlin complexes with one talin dimer (Bachir et al, 2014). Hence, as recently reported (Lu et al, 2022), talin-dependent (or tensin-dependent) dimerization of the β subunits of two active integrin heterodimers may represent the initial minimal clustering event required to allow further higher order aggregation to occur (Changede & Sheetz, 2017), finally resulting in the formation of mature and longer lived focal or fibrillar adhesions, respectively (Chastney et al, 2021).

We observed that, when incubated live on ECs, the anti-βI domain mAb 12G10, but neither the anti-hybrid domain mAb HUTS4 nor the anti-I-EGF2 domain mAb 9EG7, disrupts the aggregation of active β1 integrins into elongated fibrillar adhesions, hinting that the headpiece βI domain is mechanistically involved in active β1 integrin clustering. It is intriguing that the extracellular βI domain of integrin β subunits displays a dinucleotide-binding Rossmann fold (Xiong et al, 2001) similar to the core G domain of small GTPases, such as RAS, which must dimerize to signal (Rudack et al, 2021) and

which undergo conformational changes to bind effectors. In addition, we found that the overexpression of EGFP-tensin 1 effectively rescues the 12G10-elicited fragmentation of fibrillar adhesions. It is hence conceivable that the stabilization of the minimal clustering unit of active integrins may rely on a synergy between the dimerizing properties of cytoskeletal adaptors on the cytosolic side (Bachir et al, 2014; Changede & Sheetz, 2017; Lu et al, 2022) and integrin β subunit βI domains on the extracellular side. Our findings also prompt the hypothesis that in ECs, the βI domain–dependent interactions of active β1 integrins may be required for the formation and maintenance of fibrillar adhesions. In this context, a recent study, in which nanofiber mimetic adhesive substrates of variable size and geometry were employed, showed how lateral interactions of conformationally active integrins are central to the preservation of stable ECM adhesions and the ensuing activation of FAK, a master regulator of ECM adhesion turnover (Changede et al, 2019). Our observation that the enzymatic activity of FAK is necessary to allow 12G10 to trigger the dismantling of fibrillar adhesions suggests that interactions between active β1 integrins are physiologically counteracted by cytosolic FAK signaling, likely because of its inhibition of Rho-dependent actomyosin contractility (Ren et al, 2000; Schober et al, 2007; Tomar et al, 2009).

Live incubation of 12G10 on ECs simultaneously hampered the clustering of active β1 integrins in fibrillar adhesions and strongly increased their endocytosis. Thus, it is possible that, unless mechanically stabilized, for example, by tensin-mediated interaction with the actomyosin network, ligand-bound active β1 integrins are promptly internalized, likely via PTB domain–containing endocytic adaptors that compete with tensin for binding the membrane proximal NPxY motif of integrin β subunit cytotails (Mana et al, 2020; Elkhatib et al, 2021). From this perspective, active β1 integrin internalization and endosomal signaling (Alanko et al, 2015; Moreno-Layseca et al, 2019; Mana et al, 2020) may represent a further strategy to mechanically probe the surrounding environment. In addition to providing evidence that active β1 integrin clustering and endocytosis are oppositely regulated, our data support the concept that, albeit essential, the stabilization of β1 integrin conformational activation may not be per se sufficient to promote ECM-elicited haptotactic EC migration, which also requires clustering to both counteract the internalization of active β1 integrins from the cell surface and favor their mechanical coupling to the actin cytoskeleton.

# Materials and Methods

### Antibodies and constructs

Mouse mAbs anti-active β1 integrin clone 12G10 and clone HUTS4 and mouse mAb anti-active α5 integrin clone SNAKA51 were from

---

before bleaching. Data were then normalized on control DMSO-treated samples. Data are mean ± SD, n ≥ 31 adhesions per condition pooled from three independent experiments. Statistical analysis: unpaired *t* test, $P \leq 0.001$ ***. The effectiveness of inhibition of FAK autophosphorylation by PF-562271 was verified by Western blot analysis of EC lysates. **(E)** Representative confocal microscopy images of anti-active β1 mAb 12G10 (*green*) localization in ECs treated or not with the FAK inhibitor PF-562271; ECs were also stained for auto-phosphorylated FAK on tyrosine 397 (pFAK-Y397, *red*). Scale bar 20 μm; magnification scale bar 10 μm. Measurement of mFD of 12G10[*] adhesions revealed that, compared with control EC incubated with DMSO, 12G10[*] adhesive structures are significantly longer in presence of FAK inhibitor PF-562271. Data are mean ± SD, n ≥ 19 cells per condition pooled from three independent experiments. Statistical analysis: unpaired *t* test, $P \leq 0.0001$ ****. **(F)** Western blot analysis of FAK autophosphorylation (Y397) levels in ECs treated with or without mAb 9EG7 or mAb 12G10 for 2 or 15 min.
Source data are available for this figure.

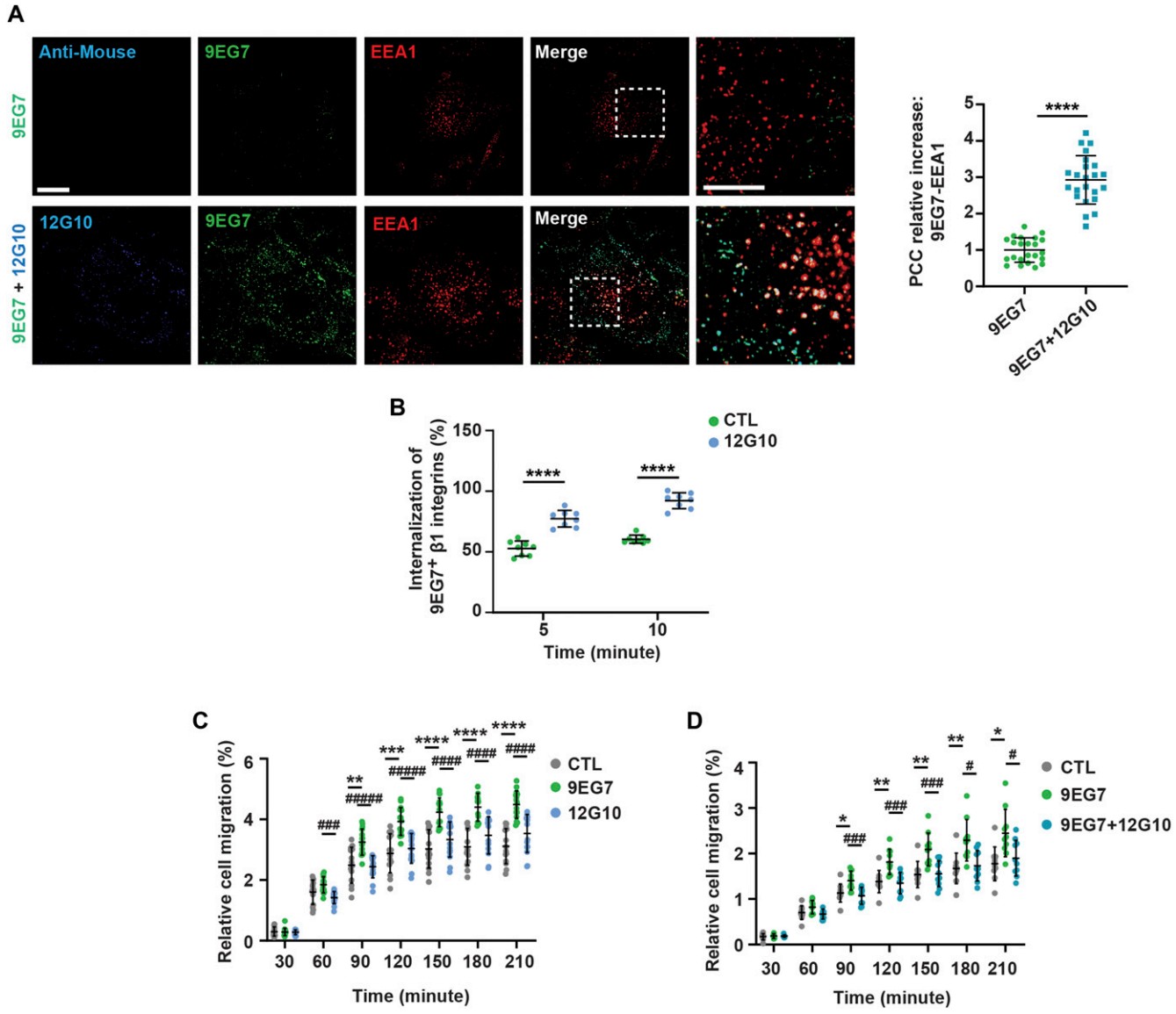

**Figure 4. Anti-βI domain mAb 12G10 promotes 9EG7⁺ active β1 integrin endocytosis and counteracts mAb 9EG7–elicited EC haptotaxis towards FN.**
**(A)** Confocal immunofluorescence microscopy analysis of ECs live treated with mAb 9EG7 (*green*) alone or in combination with mAb 12G10 (*blue*). The internalization of mAb 9EG7–bound active β1 integrins in EEA1⁺ early endosomes (*red*) was quantified by Pearson correlation coefficient (PCC). Treatment with the anti-βI domain mAb 12G10 promotes mAb 9EG7–bound active β1 integrins endocytosis. Data are mean ± SD, n ≥ 23 cells per condition pooled from three independent experiments. Scale bar 20 μm; magnification scale bar 10 μm. Statistical analysis: unpaired t test; P ≤ 0.0001 ****. **(B)** Time-course analysis of the relative amounts of endocytosed 9EG7⁺ active β1 integrins in control (CTL; *green*) versus mAb 12G10–treated (*light blue*) ECs, evaluated by internalization and capture ELISA assays. Treating living ECs with the anti-βI domain mAb 12G10 elicits a strong increase in 9EG7⁺ active β1 integrin endocytosis. Data are mean ± SD, of eight technical replicates per condition pooled from three independent experiments. Statistical analysis: two-way ANOVA and Bonferroni's post hoc analysis; P ≤ 0.0001 ****. **(C, D)** Real-time analysis of EC haptotactic migration towards FN (xCELLigence RTCA DP system) either in the absence (CTL) or the presence of anti-active β1 integrin mAb 9EG7 alone (C, D) or mAb12G10 alone (C) or combined mAb 9EG7 and mAb 12G10 (D). Data are mean ± SD, n ≥ 14 technical replicates per condition pooled from four independent experiments. Statistical analysis: two-way ANOVA and Bonferroni's post hoc analysis; P ≤ 0.05 *, #; P ≤ 0.01 **, ##; P ≤ 0.001 ***, ###; P ≤ 0.0001 ****, ####. **(C, D)** *CTL versus mAb 9EG7; # CTL versus mAb 12G10 (C); or CTL versus mAb 9EG7 + mAb 12G10 (D). Source data are available for this figure.

Merck Millipore. Unlabeled and Alexa Fluor 488–labeled rat mAb anti-active β1 integrin clone 9EG7 were from BD Biosciences. Mouse mAb anti-active β1 integrin clone TS2/16 was from BioLegend. Alexa Fluor 647–labeled mouse mAb anti-active β1 integrin clone 12G10 was from Abcam. Mouse mAb anti-β tubulin was from Sigma-Aldrich. Rabbit polyclonal Ab (pAb) anti-FAK and anti-phospho-FAK (Tyr576/577) were from Cell Signaling Technology. Goat pAb anti-EEA1 (N-19) was from Santa Cruz Biotechnology. Rabbit pAb anti-tensin 1 was from Novus Biological. Rabbit mAb anti CD61 (β3 integrin subunit) was from Invitrogen.

Fab fragments of 12G10 were produced by ficin cleavage of purified IgG followed by removal of Fc-containing fragments using protein A–Sepharose, according to the manufacturer's instructions (Thermo Fisher Scientific).

Fab fragments of 9EG7-Alexa Fluor 488 were produced by papain cleavage of purified IgG followed by removal of Fc-containing fragments using BioMag Goat anti-Rat IgG (Fc specific) (Bangs Laboratories).

HRP-conjugated secondary Abs used in Western blots were from Santa Cruz Biotechnology. Alexa Fluor–conjugated secondary Abs employed in confocal immunofluorescence were from Thermo Fisher Scientific. A FITC-tagged goat anti-mouse IgG Fab Ab (cat # F5262; Sigma-Aldrich) was employed to detect 12G10-Fab fragments in immunofluorescence analysis. Cell nuclei were labeled by far-red fluorescent carbocyanine monomer nucleic acid stain To-Pro-3 (Thermo Fisher Scientific).

pEGFP tensin1 was a gift from David Critchley and Kenneth Yamada (plasmid # 105297; http://n2t.net/addgene:105297; RRID: Addgene_105297; Addgene) (Clark et al, 2010).

### Isolation, culture, and transfection of ECs

Primary human ECs were isolated from the umbilical cords as previously described (Jaffe et al, 1973). Briefly, the umbilical vein was cannulated with a blunt, 17-gauge needle that was secured by clamping. The umbilical vein was then perfused with 50 ml of PBS to wash out the blood. Next, 10 ml of 0.2% collagenase A (Cat. # 11088793001; Roche Diagnostics) diluted in cell culture medium was infused into the umbilical vein and incubated for 30 min at RT. The collagenase solution containing the ECs was flushed from the cord by perfusion with 40 ml of PBS, collected in a sterile 50-ml centrifuge tube, and centrifuged for 5 min at 800$g$. Cells were first resuspended in M199 medium completed with cow brain extract, heparin sodium salt from porcine intestinal mucosa [0.025 mg/500 ml], penicillin/streptomycin solution, 20% FBS (Sigma-Aldrich), and subsequently plated in cell culture dishes that had been previously adsorbed with 1% gelatin from porcine skin (G9136; Sigma-Aldrich). Cells were tested for mycoplasma contamination by means of Venor GeM Mycoplasma Detection Kit (MP0025-1KT; Sigma-Aldrich) and grown at 37°C with 5% $CO_2$ in Medium 199 (Sigma-Aldrich) supplied with 20% FBS with 0.005% heparin and 0.2% brain extract (complete M199). The isolation of primary venous ECs from human umbilical cords was approved by the Office of the General Director and the Ethics Committee of the Azienda Sanitaria Ospedaliera Ordine Mauriziano di Torino hospital (protocol approval no. 586, 22 Oct 2012, and no. 26884, 28 Aug 2014), and informed consent was obtained from each patient. Primary ECs were kept in culture up to a maximum of three passages. Immunofluorescence analyses were always performed on passage 1 primary ECs.

ECs were transfected by means of Lipofectamine and PLUS reagent (Thermo Fisher Scientific).

### Immunofluorescence staining

Primary ECs (100,000 cells/coverslip) were plated in a 1:1 mix of EGM2 and Optimem, on 0.17-mm glass coverslips (no. 1.5; 12 mm diameter) that were pre-coated with 3 $\mu$g/ml human plasma FN (Cat. # 1918-FN-02M; R&D Systems) for 1 h at 37°C. The day after, cells were gently washed in PBS and fixed with a solution of 4% PFA in PBS for 7 min at RT. For live cell antibody incubation, the day after plating, anti-active $\beta$1 integrin mAbs (10 $\mu$g/ml) were directly

added to the medium, and cells were kept at 37°C for different time points depending on the assay. When necessary, upon incubation at 37°C, ECs were acid washed by a 6-min incubation at 4°C in 0.5 M NaCl, 0.5% acetic acid, pH 2.6 (acid buffer). Cells were then washed in PBS and fixed with a solution of 4% PFA in PBS for 7 min at RT.

For immunofluorescence analysis of $\beta$3 integrin in ECs, 30,000 cells/coverslip were plated on glass coverslips that were pre-coated with 5 $\mu$g/ml human vitronectin (Cat. # 2349-VN; R&D Systems) for 1 h at 37°C.

For FN incorporation into fibrils, the day after plating, cells were kept at 37°C for 1 h in the presence of soluble rhodamine-labeled FN (1.5 $\mu$g/200 $\mu$l; Cytoskeleton, Inc.). Afterwards, 12G10 mAb was directly added to the medium, and ECs were kept at 37°C for 15 min.

For experiments with 9EG7-Fab, 9EG7-Alexa Fluor 488 or its corresponding Fab was added to ECs for 5 min at 37°C. Cells were then washed and incubated either in the presence or in the absence of 12G10 mAb for 10 min at 37°C.

### Conventional confocal scanning microscopy

PFA-fixed ECs were permeabilized with 0.1% Triton X-100 in PBS for 2 min at 4°C or with 0.25% saponin, 3% BSA in PBS for 10 min at RT. Because of its ability to preserve the integrity of endosomal vesicles, permeabilization with saponin was employed in internalization assays. Primary Abs were diluted in 1% donkey serum in PBS, incubated for 1 h at RT, and revealed by appropriate fluorescently labeled secondary Abs that were diluted 1:400 1% donkey serum in PBS and incubated for 45 min at RT. Coverslips were mounted on microscope slides by using Mowiol mounting medium and allowed to dry overnight at RT.

Cells were analyzed using a Leica TCS SP8 AOBS confocal laser scanning microscope (Leica Microsystems). Fluorochromes and fluorescent proteins were excited at the optimal wavelength by means of 80 MHz pulsed white light laser (470–670 nm). For image acquisition, we used a HC PL APO CS2 63×/1.40 oil immersion objective. Image acquisition was performed by adopting the same laser power, gain, and offset settings for all the images of the same experiment and avoiding saturation. Images were analyzed using the Leica Application Suite (for colocalization analysis) or ImageJ quantification tool (for other parameters).

### Time-lapse confocal microscopy

ECs were plated in 24-well glass bottom plates #1.5H (Cod. P24-1.5H-N; Cellvis) coated with 1% gelatin from porcine skin (G9136; Merck) at a concentration of 10 × 10$^4$ cells per well. The day after seeding, ECs were placed onto a sample stage within an incubator chamber set to 37°C in an atmosphere of 5% $CO_2$, 20% humidity, and Alexa Fluor–labeled anti-active $\beta$1 integrin mAbs were added to the medium at a concentration of 10 $\mu$g/ml. ECs were imaged by using a Leica TCS SP8 AOBS confocal microscope equipped with a HC PL APO CS2 63×/1.40 oil objective and hybrid detectors. Images were recorded using a reflection-based Adaptive Focus Control for 30 min at a rate of 0.1 frame per second.

## Super-resolved time-*g*-STED confocal microscopy

Immunofluorescence stainings for *g*-STED imaging were performed as described for conventional confocal microscopy, yet fluorescently labeled secondary Abs were diluted 1:100. Leica TCS SP8 *g*-STED 3× laser scanning microscope was used to acquire super-resolved images (Leica Microsystems). A Leica STED HC PL APO 100×/1.40 objective was used. Fluorochromes and fluorescent proteins were excited at the optimal wavelength by means of 80 MHz pulsed white light laser (470–670 nm), allowing time gating of fluorescence lifetimes. For STED, the appropriate, 592 or 660 nm, depletion laser was used. 592 and 660 nm depletion lasers were used at 40% and 70% of their nominal power. Fluorescence channels were scanned sequentially, and emission was revealed by means of hybrid spectral detectors (HyD SP; Leica Microsystems). Time-gated detection was also used, and detection was delayed by 0.5 ns. Images were analyzed using ImageJ quantification tool.

## FRAP

ECs were plated in 1% porcine skin gelatin-coated Ibidi *μ*-slide eight-well chambers and oligofected with pEGFP tensin1. The day after, ECs were treated with FAK inhibitor PF-562271 (selleckchem.com) or DMSO for 30 min at 37°C and then analyzed by using a Leica TCS SP8 AOBS confocal microscope equipped with a HC PL APO CS2 63×/1.40 oil objective, a 37°C humidified and 5% CO$_2$ containing chamber, and PMTs detectors. A high-intensity 488 nm Argon laser line was used to bleach the selected ROI. Two pre-bleaches, 2 bleach, and 40 post-bleach frames were acquired at a 1.4-s interval.

## Western blot

Cells were lysed in boiling buffer (50% H$_2$O, 25% SDS 10%, and 25% Tris 1 M HCl, pH 6.8) and then sonicated (30% amplitude) for 20 s. The total protein amount was determined using the bicinchoninic acid protein assay reagent (Thermo Fisher Scientific). Equivalent amounts (50 *μ*g) of protein were resuspended in Laemmli buffer (Laemmli, 1970), resolved on a precast polyacrylamide gel (Thermo Fisher Scientific), and then transferred with Trans-Blot Turbo Mini Nitrocellulose Transfer Packs (Bio-Rad). Membranes were probed with antibodies of interest and detected by enhanced chemiluminescence (ECL; PerkinElmer).

## Integrin internalization assay

Integrin endocytosis assays were performed as previously described (Roberts et al, 2001) with minor modifications. ECs were transferred to ice, washed twice in cold PBS, and surface labeled at 4°C with 0.2 mg/ml sulfo-HS-SS-biotin (Pierce) in PBS for 30 min. Labeled cells were next washed in cold PBS and transferred to 37°C pre-warmed 10% FBS M199 medium. At different times of incubation at 37°C (5 and 10 min), the medium was removed, dishes were rapidly transferred to ice, and ECs were washed twice with ice-cold PBS. Biotin was stripped from the remaining surface proteins by incubating ECs for 1 h at 4°C with a solution containing 20 mM sodium, 2-mercaptoethanesulfonate in 50 mM Tris–HCl (pH 8.6), 100 mM NaCl. 2-Mercaptoethanesulfonate was then quenched by adding 20 mM iodoacetamide for 10 min. After two additional washes in PBS, ECs were then lysed in 25 mM Tris–HCl, pH 7.4, 100 mM NaCl, 2 mM MgCl$_2$, 1 mM Na$_3$VO$_4$, 0.5 mM EGTA, 1% Triton X-100, 5% glycerol, protease inhibitor cocktail (50 mg/ml pepstatin, 50 mg/ml leupeptin, and 10 mg/ml aprotinin; Sigma-Aldrich), and 1 mM PMSF. Afterwards, lysates were cleared by centrifugation at 12,000*g* for 20 min. Supernatants were then corrected to equivalent protein concentrations, and the amounts of internalized biotinylated integrins were determined by capture ELISA assay.

## Capture ELISA assay

Corning 96-well clear polystyrene high bind stripwell microplates (Cat #2592) were coated overnight with 5 *μ*g/ml of rat mAb anti-active *β*1 integrin clone 9EG7 in 0.05 M Na$_2$CO$_3$ (pH 9.6) at 4°C and were next blocked in PBS containing 0.05% Tween-20 (PBS-T) and 5% BSA for 1 h at RT. Internalized sulfo-NHS-SS–biotinylated 9EG7[+] active *β*1 integrins were captured by overnight incubation of 50 *μ*l EC lysates at 4°C. Unbound material was removed by extensive washing with PBS-T. Wells were then incubated with streptavidin-conjugated HRP (Amersham) in PBS-T containing 1% BSA for 1 h at 4°C. After further washing, internalized sulfo-NHS-SS–biotinylated 9EG7[+] active *β*1 integrins were detected by a chromogenic reaction with ortho-phenylenediamine that was quantified by means of a Synergy HT microplate reader (at 490 nm; BioTek Instruments) and the Gen5 software (BioTek Instruments).

## AFM on recombinant *α*5*β*1

A home-built AFM was used to measure the lifetime of recombinant integrin *α*5*β*1 and FNIII7–10 in the presence or absence of 9EG7. The AFM system and the reagents have been described previously (Kong et al, 2009). In brief, the AFM cantilever tip and a polystyrene dish were, respectively, incubated with FNIII7-10 and Fab fragments of anti-Fc mAb GG-7 overnight at 4°C and rinsed 3 times with TBS (25 mM Tris–HCl and 150 mM NaCl, pH 7.4). The dish was then incubated with *α*5*β*1-Fc for 30 min at RT and rinsed three times with TBS. To measure the bond lifetime, the dish was filled with TBS containing 1% BSA and Mn$^{2+}$. A piezoelectric translator was used to drive the dish containing the buffer with or without 9EG7 to make contact with the cantilever tip, then immediately retract a small distance (0–5 nm) to avoid non-specific binding, hold for 0.5 s to allow bond formation, and retract again to detect adhesion. If an adhesion was observed during the second retraction phase, the force was clamped at a desired level to measure the bond lifetime at constant force. This cycle was repeated a few thousand times to obtain a large number of bond lifetimes at the entire force range, which were segregated into several force bins and shown as mean ± SEM bond lifetime versus force.

## Migration assay

Real-time directional migration of ECs towards FN was monitored with an xCELLigence RTCA DP instrument (ACEA Biosciences). The bottom side of the upper chamber (the side facing the lower chamber) of CIM-Plate 16 was coated with 30 *μ*l of 3 *μ*g/ml FN for 30 min at RT. Each lower chamber well was first filled with 160 *μ*l of

complete M199 and then assembled to the upper chamber. Each upper chamber well was then filled with 30 $\mu$l of complete M199. The plate was put for 1 h at 37°C. The experiment file was set up using the RTCA Software 1.2. ECs were detached and resuspended to a final concentration of 30 × $10^3$ cells/100 $\mu$l. The BLANCK step was started to measure the background impedance of cell culture medium, which was then used as reference impedance for calculating CI values. 100 $\mu$l of cell suspension (30 × $10^3$ cells) was briefly (5 min) incubated on ice with or without (CTL) the mAbs and then added to each well of the upper chamber. The CIM-Plate 16 was placed in the RTCA DP instrument equilibrated in a $CO_2$ incubator. EC migration was continuously monitored, and cell index was measured every 10 min using the RTCA DP instrument. Average, SD, and *P*-value were calculated on the CI data exported from RTCA instrument for the technical replicates of each experimental condition over time. Migration data are represented as a percentage considering the control samples as 100%.

### Statistical analysis

For statistical evaluation of in vitro experiments, data distribution was assumed to be normal, but this was not formally tested. Parametric two-tailed heteroscedastic *t* test was used to assess the statistical significance when two groups of unpaired normally distributed values were compared; when more than two groups were compared, parametric one-way or two-way ANOVA with Bonferroni's post hoc analysis was applied. For all quantifications, SD is shown. All data were analyzed with Prism software (GraphPad Software).

# Data Availability

Raw data of uncropped scans of Western blots and all graphs are available as source data.

# Supplementary Information

# Acknowledgements

We are grateful to Giulia Villari for critically reading the article. The research leading to these results has received funding from AIRC under IG 2018, ID. 21315—P.I. G Serini and IG 2017, ID. 20366—P.I. D Valdembri; Ministero dell'Istruzione, dell'Università e della Ricerca (PRIN 2020EK82R5), P.I. G Serini; Università di Torino, Bando Ricerca Locale 2019 (CUP D84I19002940005), P.I. G Serini; Cancer Research UK, grants C13329 and DRCRPG-May21\100002—P.I. MJ Humphries; BBSRC grant BB/P000681/1—P.I. C Ballestrem; NIH grant R01HL132019—P.I. C Zhu.

### Author Contributions

G Mana: conceptualization, data curation, formal analysis, validation, investigation, visualization, methodology, and writing—original draft, review, and editing.

D Valdembri: conceptualization, data curation, formal analysis, supervision, funding acquisition, validation, investigation, visualization, methodology, and writing—original draft, review, and editing.

JA Askari: resources, investigation, and methodology.

Z Li: resources, data curation, formal analysis, funding acquisition, validation, investigation, visualization, and methodology.

P Caswell: conceptualization, resources, supervision, validation, visualization, methodology, and writing—original draft, review, and editing.

C Zhu: conceptualization, supervision, funding acquisition, visualization, methodology, and writing—original draft, review, and editing.

MJ Humphries: conceptualization, resources, supervision, funding acquisition, visualization, and writing—original draft, review, and editing.

C Ballestrem: resources, supervision, funding acquisition, investigation, visualization, and writing—original draft, review, and editing.

G Serini: conceptualization, formal analysis, supervision, funding acquisition, visualization, and writing—original draft, review, and editing.

### Conflict of Interest Statement

The authors declare that they have no conflicts of interest.

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
