## [Reviewer comments · Life Science Alliance]

Life Science Alliance

The β I domain promotes active β 1 integrin clustering into mature adhesion sites

Guido Serini, Giulia Mana, Donatella Valdembri, Janet Askari, Zhenhai Li, Patrick Caswell, Cheng Zhu, Martin Humphries, and Christoph Ballestrem

DOI: <https://doi.org/10.26508/lsa.202201388>

Corresponding author(s): Guido Serini, University of Turin

Review Timeline:

Submission Date:	2022-01-25
Editorial Decision:	2022-02-16
Revision Received:	2022-10-10
Editorial Decision:	2022-10-28
Revision Received:	2022-11-02
Accepted:	2022-11-03

Scientific Editor: Novella Guidi

Transaction Report:

February 16, 2022

Re: Life Science Alliance manuscript #LSA-2022-01388-T

Prof. Guido Serini
University of Torino
Department of Oncology
Candiolo Cancer Institute- FPO-IRCCS
Candiolo, Torino 10060
Italy

Dear Dr. Serini,

Thank you for submitting your manuscript entitled "The β 1 domain promotes active β 1 integrin clustering into mature adhesion sites" to Life Science Alliance. The manuscript was assessed by expert reviewers, whose comments are appended to this letter. As you will note from the reviewers' comments below, all reviewers are quite interested and think that this manuscript provides a significant advance. However, they do raise several concerns that would need to be addressed in the revised version, with the main one being the lack of a molecular mechanism that explains why one of the antibodies causes disassembly of fibrillar adhesions. Specifically, how 12G10 mediates fibrillar adhesions (FBs) formation-inhibiting effect. As suggested by Reviewer 1, please address this point by performing binding assays (MST) or crosslinking experiments and test the stabilizing effect of 9EG7 with Fab fragment to exclude that the effect is solely due to the bivalency of the antibody. Another key experiment is to examine whether 12G10 alters fibronectin assembly and compare it with 9EG7 to test whether the antibodies are altering ligand binding differently. All the other concerns raised by the reviewers should be addressed as well. We, thus, encourage you to submit a revised version of the manuscript back to LSA that responds to all of the reviewers' points.

Thank you for this interesting contribution to Life Science Alliance. We are looking forward to receiving your revised manuscript.

Sincerely,

-- Summary blurb (enter in submission system): A short text summarizing in a single sentence the study (max. 200 characters including spaces). This text is used in conjunction with the titles of papers, hence should be informative and complementary to

the title and running title. It should describe the context and significance of the findings for a general readership; it should be written in the present tense and refer to the work in the third person. Author names should not be mentioned.

B. MANUSCRIPT ORGANIZATION AND FORMATTING:

Reviewer #1 (Comments to the Authors (Required)):

Serini and coworkers show in this interesting manuscript that the b1 domain promotes clustering of activated b1 integrins. To reveal this finding the authors use three antibodies that bind to and stabilize distinct conformations of b1 integrins: 12G10 that binds the b1 domain and HUTS4 that bind the hybrid domain and 9EG7 that bind the second EGF-like domain of b1 integrins. 12G10 and HUTS4 recognize and stabilize the extended open conformation and 9EG7 both the extended closed and extended open conformations of b1 integrins. Immunostaining of FN-seeded endothelial cells with these antibodies identified similarly sized and distributed fibrillar adhesions (FBs), while incubation of live cells revealed that 12G10 but not the other two antibodies diminished the formation of FBs. Experiments presented in the manuscript also demonstrate that preincubation of 9EG7, overexpression of tension and chemical inhibition of the catalytic activity of FAK prevent the 12G10-induced impact on FB formation, endocytosis of active b1 integrins and cell migration.

Altogether, these are unexpected and novel findings and will be highly interesting for the adhesion community. The paper is also excellently written and easy to read. The only downer is that the authors did not uncover how 12G10 mediates FB formation-inhibiting effect. Interestingly, Serini and colleagues discuss a potential involvement of the b1 domain containing Rossman fold for potentially dimerizing b1 integrins. Since the authors have ectodomains (used in the AFM experiments) they may want to confirm or exclude this possibility with binding assays (MST) or crosslinking experiments. Finally, the stabilizing effect of 9EG7 should be tested with Fab fragments to exclude that the effect is solely due to the bivalency of the antibody.

Reinhard Fässleer

Reviewer #2 (Comments to the Authors (Required)):

This paper starts with an intriguing result; incubating live cells with three different monoclonal antibodies that recognise active conformations of beta-containing integrin heterodimers gives different results: one MAb results in the disassembly of fibrillar adhesions, whereas the other two do not. Additional experiments demonstrate that a number of different manipulations can block the MAb-induced disassembly. Overall the experiments are performed well and documented clearly. The paper falls short of providing a molecular mechanism that explains why one of the antibodies causes disassembly, and in my view draws conclusions that are not supported by the findings; I can certainly think of alternative interpretations of the data, as elaborated below. Furthermore, the lack of a clear model to explain these results makes it difficult to see how this paper would be of interest to those outside the field.

1. Major Issue: Interpretation. The authors show that the application of 12G10 MAb induces disassembly of fibrillar adhesions and endocytosis of integrins. As this MAb binds to the most extracellular position on the integrin, i.e. farthest from the plasma membrane and closest to the ligand, a simple explanation to explain this result is that while it does bind and stabilize the active conformation of the integrin, perhaps it results in detachment of the integrin from the fibronectin fibrils, permitting integrin endocytosis. This can be countered by treatments that stimulate the activity of the integrin: 1) binding of one of the other MAbs, 9EG7, 2) overexpression of tensin (also known to inside-out activate integrins), 3) inhibition of the negative regulator of integrins, FAK. I am somewhat perplexed as to why the authors interpret these experiments to conclude that MAb 12G10 is altering clustering of integrins. I cannot see any experiments that directly address integrin clustering. The only counter argument I can see to the above interpretation (detachment from ligand) is that reduced binding to ligand is apparently not evident in the AFM experiments, as 9EG7 and 12G10 may have similar effects. However, I note that in the Kong paper, a second MAb similar to 12G10, TS2/16, stimulates much more persistent binding; perhaps this does indicate that 12G10 reduces ligand interaction. In addition, there may be differences in accessibility of the integrin binding site within intact FN fibrils versus the AFM setup. That fact that 9EG7 can suppress the effect of 12G10 suggests that it is not steric hindrance, but that 12G10 converts integrin into a

conformation with reduced affinity for ligand, which 9EG7 can block. This scenario seems more feasible than an unspecified molecular mechanism that alters clustering.

The above points highlight two clear omissions in this manuscript: 1. If the aim of the AFM experiment is to show a comparison between two antibodies it is necessary to show the data for both; its not reasonable to require that the reader look up the previous work for comparison. 2. It is essential to compare the affect of 12G10 application to live cells with application of additional antibodies that bind to the betal domain, in particular TS2/16, and other ones available. At the moment it is not possible to work out if it is the region of binding or the particular epitope that is important (in addition, providing detailed epitope information would be helpful for the reader). I find this alternative interpretation just as interesting as a potential effect on clustering, as it may give additional insights into the conformational changes that accompany integrin binding and its catch bond behaviour.

Minor issues.

2. Figure 1 B and C, add label to figure so that difference between experiments is clear without having to read the legend.
3. p 6 end of first paragraph. This concluding sentence is overinterpreted in a couple of ways. There is insufficient data to conclude that it is the domain to which each antibody binds that is the key difference in their effect, thus the transfer of the conclusion to the domains is not proven. Second, there is no evidence that aggregation of integrins is being altered- as later data shows, the integrins are being endocytosed, which disupts the fibrillar adhesion. I cannot see a reason to invoke aggregation.
4. p6 first sentence of 2nd paragraph. The wording is incorrect, because it implies we have observed 9EG7 positive adhesions being disassembled by the addition of 12G10, when it fact, the data shows that staining cells preincubated for 10 minutes with 12G10 with 9EG7 shows the active integrins are now dispersed. In fact, as is later shown, it is not possible to disaggregate 9EG7-stained fibrillar adhesions by addition of 12G10.
5. p6 third sentence. Again a wording issue. It is contradictory "to verify" something doesnt work in an alternative way. To test whether.. would be better wording.
6. p 7 Suppl. Video 3 shows an interesting experiment, live imaging cells upon the addition of labelled 12E10. The cell in the middle seems to lack any fibrillar adhesions, but the cells surrounding them do have them, and its possible one can see them disperse-its hard with the changing focal plane. This key movie could be improved, and it would be helpful to quantify the disassembly process. It would also be very informative to do the experiment with GFP-tagged integrin- is all integrin endocytosed or just that bound by 12E10?
6. p8 first paragraph, last sentence. There is no compelling evidence that I know of, or that has been presented, that either tensin expression or 9EG7 binding stabilize clustering, whereas there is evidence that they increase integrin binding to ligand. Therefore, the sentence should more accurately be written as ... can be prevented by increasing integrin activity from either the outside or inside of the cell.
7. p8 last line, I think arguing that disassembly requires permissive steady activity of FAK is an over statement. As mentioned above, inhibiting FAK, and adding more tensin and adding 9EG7 all elevate integrin activity. To me the data argues that when integrin is activated in certain ways, 12G10 is not able to detach the integrin from ligand and permit endocytosis. It would be interesting to discover why this is the case. That is why finding out if it is a unique feature of this betal domain-binding antibody or shared with others will be informative.
8. p9. Last experiment. Since 12E10 removes the integrin from the surface, is it surprising that it impairs integrin dependent migration on an ECM? This data does not support the assertion that this provides evidence for the importance of clustering.
9. p11. Sentence starting Our major finding... I disagree with this conclusion. The fact that one "activating" antibody removes integrin from the surface does not address whether activation is sufficient or not.
10. This sentence is an overstatement "Our observation that the enzymatic activity of FAK is necessary to allow 12G10 to trigger the dismantling of fibrillar adhesions suggests that extracellular β 1 domain-dependent interactions between active β 1 integrins are physiologically counteracted by cytosolic FAK signaling..." No evidence has been presented that 12E10 blocks "extracellular β 1 domain-dependent interactions between active β 1 integrins"

Referee Cross-Comments

I am curious to know the other reviewers thoughts on the alternative intepretation I put forward.

The experiments suggested by Reviewer 1 would indeed provide some evidence to support the hypothesis that 12G10 is having its specific effect by blocking interactions between integrins (clustering).

The request by reviewer 3 to examine whether 12G10 alters fibronectin assembly is an excellent suggestion, with the addition that it would be valuable to compare with 9EG7- this would test whether the antibodies are altering ligand binding differently. It is not clear to me why reviewer 3 has asked for more experiments proving that it is alpha5 in the heterodimers being involved. They raise a fair point, but I think it would be simpler to just remove the the labelling of the alpha subunit as alpha5 from Figure 1. Its not clear to me that knowing the identity of the alpha subunit will alter the intepretation of the experiments.

Reviewer #3 (Comments to the Authors (Required)):

The MS by Mana et al is a careful study of the role of different beta1 integrin domains in integrin conformational changes and integrin clustering in fibrillar adhesion formation and cell migration. The authors use three different beta1 integrin mAbs, with well-defined epitopes, to demonstrate that the formation of fibrillar adhesions depends on beta1 domain and integrin clustering. The importance of FAK for integrin turnover in fibrillar adhesions is also demonstrated.

I have no comments as to experimental design or figures with excellent figure legends but have the following comments.

- The authors in figure one show alpha5beta1 and mention its role in fibrillar adhesion formation. Throughout the study only beta1 integrin antibodies are used.

- The authors use EC throughout the MS to study fibrillar adhesion formation. In vivo endothelial cells sit on a basement membrane and under physiological conditions probably only to a limited degree are involved in fibrillar adhesion formation.

1. For future studies, using other cell types more traditionally involved in fibrillar adhesion formation, like fibroblasts, it would be useful to know what the set-up of beta1 integrins are in the cells studied. More specifically, what other fibronectin-binding beta1 integrins do these cells express?

2. The authors use beta1 integrin antibodies to determine the domains in beta1 chain in the alpha5beta1 heterodimer which is involved in fibrillar adhesion formation.

- The authors should also perform staining with alpha5 antibody, like SNAKA51, to confirm staining pattern.

- In the discussion the authors should mention recent results in superresolution microscopy suggesting that fibrillar adhesions are composed of nanodomains (1). Is there a difference between alpha5 and beta1 integrin immunostaining in this regard?

- Immunostaining for beta3 integrin would be informative since it has been reported to localize at anchoring points of fibrillar adhesions; is this localization retained when fibrillar adhesion integrity is disturbed with beta1 integrin antibodies?

- Do treatment of cells with 12G10 also affect fibronectin assembly?

Reference cited:

(1) Tomer, D. et al. New mechanism of of fibronectin fibril assembly revealed by live imaging and super-resolution microscopy. bioRxiv <https://doi.org/10.1101/2020.09.09.290130> (2020).

Reviewer #1 (Prof. Reinhard Fässler)

Serini and coworkers show in this interesting manuscript that the $\beta 1$ domain promotes clustering of activated $\beta 1$ integrins. To reveal this finding the authors use three antibodies that bind to and stabilize distinct conformations of $\beta 1$ integrins: 12G10 that binds the $\beta 1$ domain and HUTS4 that bind the hybrid domain and 9EG7 that bind the second EGF-like domain of $\beta 1$ integrins. 12G10 and HUTS4 recognize and stabilize the extended open conformation and 9EG7 both the extended closed and extended open conformations of $\beta 1$ integrins. Immunostaining of FN-seeded endothelial cells with these antibodies identified similarly sized and distributed fibrillar adhesions (FBs), while incubation of live cells revealed that 12G10 but not the other two antibodies diminished the formation of FBs. Experiments presented in the manuscript also demonstrate that preincubation of 9EG7, overexpression of tension and chemical inhibition of the catalytic activity of FAK prevent the 12G10-induced impact on FB formation, endocytosis of active $\beta 1$ integrins and cell migration.

Altogether, these are unexpected and novel findings and will be highly interesting for the adhesion community. The paper is also excellently written and easy to read. The only downer is that the authors did not uncover how 12G10 mediates FB formation-inhibiting effect. Interestingly, Serini and colleagues discuss a potential involvement of the $\beta 1$ domain containing Rossman fold for potentially dimerizing $\beta 1$ integrins.

1. Since the authors have ectodomains (used in the AFM experiments) they may want to confirm or exclude this possibility with binding assays (MST) or crosslinking experiments.

As suggested by Prof. Fässler, together with the Biomolecular Analysis Facility of the Wellcome Trust Centre for Cell-Matrix Research in Manchester, we worked to set up experiments aimed at formally testing the *in vitro* clustering of $\alpha 5\beta 1$ integrins and its possible modulation by 12G10 or 9EG7 mAbs, by employing a **microscale thermophoresis (MST)** and a purified $\alpha 5\beta 1$ integrin. Unfortunately, as explained by the experts of the Facility, **MST can not be used to assess homomeric interactions**, as the labelled/unlabelled combinations would be all interacting, making the analyses confusing and not allowing to get clear-cut results.

To try circumventing the MST issue, we sought to exploit a **Proximity Ligation Assay (PLA)** on endothelial cells to assess active $\beta 1$ integrin clustering with **HUTS-4 antibody that we found not to influence clustering**. To this aim, by employing the Duolink Proximity Ligation Assays, we labelled HUTS-4 with either the MINUS or the PLUS oligonucleotide probe and used a 50:50 mix of HUTS-4-MINUS and HUTS-4-PLUS to quantify active $\beta 1$ integrin clustering. **To make sure that we did not detect the clustering events outside adhesion sites, we modified the PLA assay**. Live ECs were pre-incubated for 5 min with the 50:50 HUTS-4-MINUS/HUTS-4-PLUS mix (to allow its binding to active $\beta 1$ integrin within adhesion sites) or HUTS-4-PLUS only (as negative control), washed with PBS to remove the unbound excess probe, and then incubated or not with 12G10 for 10 min. Next, ECs were fixed with 2% PAF for 5 min and the PLA assay was then executed following the standard protocol. HUTS-4⁺ active $\beta 1$ integrin clustering was then quantified by measuring the total number of PLA dots per cell either in the presence or the absence of 12G10. As a control, we also performed the experiment using HUTS-4-PLUS only to rule out PLA non-specific effects. **We found that 12G10 significantly reduces HUTS-4⁺ active $\beta 1$ integrin clustering in ECs (Fig. for Reviewer #1 below)**. Since we had to modify the PLA protocol to make it suitable to our scientific question, **we are providing our data for the Reviewer, but we would not include them in the manuscript** as we think it would need further validation to exclude possible interferences introduced by those modifications.

2. Finally, the stabilizing effect of 9EG7 should be tested with Fab fragments to exclude that the effect is solely due to the bivalency of the antibody.

We produced the **Fab fragment of 9EG7** and compared its ability of disrupting 12G10 effects on active $\beta 1$ integrin clustering in fibrillar adhesions of living ECs to the one exerted by 9EG7 intact form. The experiments clearly

showed that **only the pre-incubation with the intact 9EG7 mAb, but not its Fab, is able to neutralize the fragmenting activity that 12G10 exerts** on active $\beta 1$ integrin⁺ fibrillar adhesions of living ECs (Page 9 and Suppl Fig 3). This implies that **the effect of intact 9EG7 mAb** is not simply **due to** its ability to stabilize the conformation of active $\beta 1$ integrins, but also to **its clustering activity** that crucially relies on its dimeric nature.

Figure for Reviewer #1. 12G10 significantly reduces HUTS-4⁺ active $\beta 1$ integrin clustering in endothelial cells as revealed by Proximity Ligation Assay (PLA). The clustering of HUTS-4⁺ $\beta 1$ integrins was detected by using use of Duolink[®] PLA reagents (Sigma Aldrich). Immunofluorescent PLA signals are shown in *green* and the DRAQ5-labeled nuclei in *gray*. *Top left*, negative control (CTL-), where HUTS-4 was combined to PLUS probe only. *Middle left*, basal $\beta 1$ integrin clustering (CTL+), where HUTS-4 was combined to both PLUS and MINUS probes. *Bottom left*, impact of 12G10 on $\beta 1$ integrin clustering, as detected by HUTS-4 mAb (in the presence of both PLUS and MINUS probes). *Right graph*, data are mean \pm SD. For both CTL+ and 12G10 treated endothelial cells, $n \geq 27$ images per condition from 3 independent experiments. Negative control (CTL-), $n = 10$ images from 2 independent experiments. Statistical analysis: one-way ANOVA and Bonferroni's post hoc analysis; $p \leq 0,0001$ ****.

Reviewer #2

This paper starts with an intriguing result; incubating live cells with three different monoclonal antibodies that recognise active conformations of beta-containing integrin heterodimers gives different results: one MAb results in the disassembly of fibrillar adhesions, whereas the other two do not. Additional experiments demonstrate that a number of different manipulations can block the MAb-induced disassembly. Overall the experiments are performed well and documented clearly. The paper falls short of providing a molecular mechanism that explains why one of the antibodies causes disassembly, and in my view draws conclusions that are not supported by the findings; I can certainly think of alternative interpretations of the data, as elaborated below. Furthermore, the lack of a clear model to explain these results makes it difficult to see how this paper would be of interest to those outside the field.

1A. Major Issue: Interpretation. The authors show that the application of 12G10 MAb induces disassembly of fibrillar adhesions and endocytosis of integrins. As this MAb binds to the most extracellular position on the integrin, i.e. farthest from the plasma membrane and closest to the ligand, a simple explanation to explain this result is that while it does bind and stabilize the active conformation of the integrin, perhaps it results in detachment of the integrin from the fibronectin fibrils, permitting integrin endocytosis. This can be countered by treatments that stimulate the activity of the integrin: 1) binding of one of the other MAbs, 9EG7, 2) overexpression of tensin (also known to inside-out activate integrins), 3) inhibition of the negative regulator of integrins, FAK. I am somewhat perplexed as to why the authors interpret these experiments to conclude that MAb 12G10 is altering clustering of integrins. I cannot see any experiments that directly address integrin clustering. The only counter argument I can see to the above interpretation (detachment from ligand) is that reduced binding to ligand is apparently not evident in the AFM experiments, as 9EG7 and 12G10 may have similar effects. However, I note that in the Kong paper, a second MAb similar to 12G10, TS2/16, stimulates much more persistent binding; perhaps this does indicate that 12G10 reduces ligand interaction. In addition, there may be differences in accessibility of the integrin binding site within intact FN fibrils versus the AFM setup. That fact that 9EG7 can suppress the effect of 12G10 suggests that it is not steric hindrance, but that 12G10 converts integrin into a conformation with reduced affinity for ligand, which 9EG7 can block. This scenario seems more feasible than an unspecified molecular mechanism that alters clustering.

The laboratory of one of the authors, Prof. Martin J. Humphries, previously showed that **12G10 significantly increases rather than disrupting $\alpha 5 \beta 1$ binding to fibronectin** (Mould et al., *FEBS Lett.*, 1995, 363:118-122). Therefore, the disrupting effects that 12G10 exerts on the clustering of active $\beta 1$ integrin in fibrillar adhesions of living ECs cannot depend on the ability of 12G10 to reduce the interaction of active $\beta 1$ integrins with ligands, such as fibronectin.

Furthermore, we produced the **Fab fragment of 9EG7** and compared the ability of **intact 9EG7 mAb** and its Fab fragment to counteract the disrupting effects of 12G10 on active $\beta 1$ integrin clustering in fibrillar adhesions of living ECs. The experiments clearly showed that **only the pre-incubation with the intact 9EG7 mAb, but not its Fab, is able to neutralize the fragmenting activity that 12G10 exerts** on active $\beta 1$ integrin⁺ fibrillar adhesions of living ECs (**Page 9 and Suppl Fig 3**). This implies that **the effect of intact 9EG7 mAb** is not simply **due to** its ability to stabilize the conformation of active $\beta 1$ integrins, but also to **its clustering activity** that crucially relies on its dimeric nature.

1B. The above points highlight two clear omissions in this manuscript: A. If the aim of the AFM experiment is to show a comparison between two antibodies it is necessary to show the data for both; its not reasonable to require that the reader look up the previous work for comparison.

The aim of the AFM experiments was to assess, for the first time, whether 9EG7 may shift the $\alpha 5\beta 1$ integrin-FN catch bond to a lower force range, as 12G10 had already been shown to do by Prof. Cheng Zhu (Kong et al., *J. Cell Biol.*, 2009). Those AFM experiments with 9EG7 were performed some time ago by Dr. Zenhai Li in Prof. Cheng Zhu lab (Georgia Institute of Technology, Atlanta, GA). Unfortunately, nowadays, neither Dr. Zhu lab, nor Dr. Li lab, now based at Shanghai University, dispose of an AFM machinery to be employed for supplementary experiments. Therefore, even if we would really like to, we are yet unable in practice to repeat this set of AFM experiments, directly comparing 12G10 and 9EG7 ability to shift the $\alpha 5\beta 1$ integrin-FN catch bond to a lower force range.

1C. The above points highlight two clear omissions in this manuscript: B. It is essential to compare the affect of 12G10 application to live cells with application of additional antibodies that bind to the beta I domain, in particular TS2/16, and other ones available. At the moment it is not possible to work out if it is the region of binding or the particular epitope that is important (in addition, providing detailed epitope information would be helpful for the reader). I find this alternative interpretation just as interesting as a potential effect on clustering, as it may give additional insights into the conformational changes that accompany integrin binding and its catch bond behaviour.

We had already tested TS2/16, yet we did not include these data in the first version of our manuscript. We have now incorporated the TS2/16 data in Figs. 1E and 2A, showing that, similarly to 12G10, also TS2/16 reduces the maximum Feret's diameter (mFD) of elongated 9EG7⁺ active $\beta 1$ integrin (Page 6 and Fig. 1E) and tensin 1⁺ (Page 8 and Fig. 2A) clusters.

As mentioned above (point #1A), the laboratory of one of the authors, Prof. Martin J. Humphries, previously showed that both 12G10 and TS2/16 significantly increase rather than disrupting $\alpha 5\beta 1$ binding to fibronectin (Mould et al., *FEBS Lett.*, 1995, 363:118-122). Therefore, the disrupting effects that 12G10 and TS2/16 exert on the clustering of active $\beta 1$ integrin in fibrillar adhesions of living ECs cannot depend on the ability of 12G10 and TS2/16 to reduce the interaction of active $\beta 1$ integrins with ligands, such as fibronectin.

Minor issues:

2. Figure 1 B and C, add label to figure so that difference between experiments is clear without having to read the legend.

We labeled panel B and panel C of Figure 1 as "Fixed ECs" and "Living ECs", respectively.

3. p 6 end of first paragraph. This concluding sentence is overinterpreted in a couple of ways. There is insufficient data to conclude that it is the domain to which each antibody binds that is the key difference in their effect, thus the transfer of the conclusion to the domains is not proven. Second, there is no evidence that aggregation of integrins is being altered- as later data shows, the integrins are being endocytosed, which disrupts the fibrillar adhesion. I cannot see a reason to invoke aggregation.

The impact that all the mAbs employed in our study (12G10, TS2/16, HUTS-4, and 9EG7) have on the structure of $\alpha 5\beta 1$ integrin has been carefully analyzed and characterized in a relatively recent article from the lab of Timothy Springer (**Su et al., 2016, Proc Natl Acad Sci USA 113: E3872-81**) and co-authored by one of us (Prof. Martin J. Humphries). As shown in Su et al. (2016), all these mAbs specifically **recognize and stabilize the conformation of $\alpha 5\beta 1$ integrin domains in which the respective activation-dependent epitopes are located**.

As mentioned above (point #1A), we produced the **Fab fragment of 9EG7** and compared its ability of disrupting 12G10 effects on active $\beta 1$ integrin clustering in fibrillar adhesions of living ECs to the one exerted by 9EG7 intact form. The experiments clearly showed that **only the pre-incubation with the intact 9EG7 mAb, but not its Fab, is able to neutralize the fragmenting activity that 12G10 exerts** on active $\beta 1$ integrin⁺ fibrillar adhesions of living ECs (**Page 9 and Suppl Fig 3**). This implies that **the effect of intact 9EG7 mAb** is not simply **due to** its ability to stabilize the conformation of active $\beta 1$ integrins, but also to **its clustering activity** that crucially relies on its dimeric nature.

In the concluding sentence (**now at the end of page 6**), we substituted the verb “indicate” with the verb “**suggest**”.

4. p6 first sentence of 2nd paragraph. The wording is incorrect, because it implies we have observed 9EG7 positive adhesions being disassembled by the addition of 12G10, when in fact, the data shows that staining cells preincubated for 10 minutes with 12G10 with 9EG7 shows the active integrins are now dispersed. In fact, as is later shown, it is not possible to disaggregate 9EG7-stained fibrillar adhesions by addition of 12G10.

That sentence aims at describing the data in **Fig. 1E and Suppl. Fig. 1, showing that the co-incubation of 12G10 and 9EG7**, results in elongated **active $\beta 1$ integrin clusters whose** maximum Feret’s diameter (**mFD**) is **significantly shorter than the one measured in the absence of 12G10**. In other words, not only pre-incubation, but also co-incubation of the two Abs, disrupts the organization of 9EG7⁺ active $\beta 1$ integrins in elongated fibrillar adhesions.

5. p6 third sentence. Again, a wording issue. It is contradictory "to verify" something doesn't work in an alternative way. To test whether... would be better wording.

On page 7 last paragraph, we modified the wording from “verify” to “**test**”.

6. p 7 Suppl. Video 3 shows an interesting experiment, live imaging cells upon the addition of labelled 12E10. The cell in the middle seems to lack any fibrillar adhesions, but the cells surrounding them do have them, and it's possible one can see them disperse-it's hard with the changing focal plane. This key movie could be improved, and it would be helpful to quantify the disassembly process. It would also be very informative to do the experiment with GFP-tagged integrin- is all integrin endocytosed or just that bound by 12E10?

It is not possible to quantify the disassembling process by simply looking at 12G10 because we would have no control to refer to (since **12G10 destroys fibrillar adhesions** of living cells), but **an equivalent quantification was done by analyzing 9EG7 and tensin 1 in cells that were incubated or not with 12G10**.

β1-GFP cannot be employed because when transfected in β1 integrin expressing **endothelial cells**, likely due to quantity control mechanisms (Hegde and Ploegh, 2010, Curr Opin Cell Biol, 22:437-446), **β1-GFP remains stuck in the endoplasmic reticulum and does not reach the cell surface.**

Movies were performed in a **Leica SP8 confocal microscope** equipped with a **thermostatic 37 °C incubator** and laser-based **“Perfect Focus” mechanism** for time-lapse imaging. We repeated experiments, but, in our hands, it has been impossible to obtain a perfectly stable focal plane for all the cells present in such a wide field of view for a 30 min-long movie. We would also like to emphasize that **the evident absence of fibrillar adhesions in 12G10-treated living endothelial cells** is confirmed by confocal images of post-fixed cells in **Figs. 1D, 2A, 3E** and **Suppl. Fig. 1.**

7. p8 first paragraph, last sentence. There is no compelling evidence that I know of, or that has been presented, that either tensin expression or 9EG7 binding stabilize clustering, whereas there is evidence that they increase integrin binding to ligand. Therefore, the sentence should more accurately be written as ... can be prevented by increasing integrin activity from either the outside or inside of the cell.

On page 9, first paragraph, we rephrased the sentence as requested by the Reviewer.

8. p8 last line, I think arguing that disassembly requires permissive steady activity of FAK is an over statement. As mentioned above, inhibiting FAK, and adding more tensin and adding 9EG7 all elevate integrin activity. To me the data argues that when integrin is activated in certain ways, 12G10 is not able to detach the integrin from ligand and permit endocytosis. It would be interesting to discover why this is the case. That is why finding out if it is a unique feature of this β1 domain-binding antibody or shared with others will be informative.

12G10 also conformationally activates β1 integrins (e.g. see Su et al., 2016, Proc Natl Acad Sci USA 113: E3872-81). Therefore, it seems contradictory to assume that β1 integrin activation may in turn block the disassembling action of 12G10.

9. p9. Last experiment. Since 12E10 removes the integrin from the surface, is it surprising that it impairs integrin dependent migration on an ECM? This data does not support the assertion that this provides evidence for the importance of clustering.

As mentioned above, (point #1A and #3), our new experiments clearly showed that **only pre-incubation with the intact 9EG7 mAb, but not its Fab, is able to neutralize the fragmenting activity that 12G10 exerts** on active β1 integrin⁺ fibrillar adhesions of living ECs (**Page 9 and Suppl Fig. 3**). This implies that **the effect of intact 9EG7 mAb** is not simply **due to** its ability to stabilize the conformation of active β1 integrins, but also to **its clustering activity** that crucially relies on its dimeric nature. In this scenario, the internalization of β1 integrins follows and does not precede the disassembly of fibrillar adhesions.

10. p11. Sentence starting *Our major finding...* I disagree with this conclusion. The fact that one "activating" antibody removes integrin from the surface does not address whether activation is sufficient or not.

Since it is well established that both 12G10 and 9EG7 mAb conformationally activate $\beta 1$ integrin (e.g., see Su et al., 2016, Proc Natl Acad Sci USA 113: E3872-81), if it were simple activation that drove the processes of $\beta 1$ integrin-dependent adhesion formation and migration, then these two mAbs should have the same effects when incubated on living cells. Instead, they do not. Thus, this implies that in addition to conformational activation, there is another crucial mechanism by which 12G10 interferes. As mentioned above, (point #1A, #3, and #9), our new experiments clearly showed that **only pre-incubation with the intact 9EG7 mAb, but not its Fab, is able to neutralize the fragmenting activity that 12G10 exerts** on active $\beta 1$ integrin⁺ fibrillar adhesions of living ECs (**Page 9 and Suppl. Fig. 3**). This implies that **this effect of intact 9EG7 mAb** is not simply **due to** its ability to stabilize the conformation of active $\beta 1$ integrins, but also to **its clustering activity** that crucially relies on its dimeric nature. In this scenario, the internalization of $\beta 1$ integrins follows and does not precede the disassembly of fibrillar adhesions.

11. This sentence is an overstatement *"Our observation that the enzymatic activity of FAK is necessary to allow 12G10 to trigger the dismantling of fibrillar adhesions suggests that extracellular $\beta 1$ domain-dependent interactions between active $\beta 1$ integrins are physiologically counteracted by cytosolic FAK signaling..."* No evidence has been presented that 12E10 blocks "extracellular $\beta 1$ domain-dependent interactions between active $\beta 1$ integrins"

On page 13, we removed "extracellular $\beta 1$ domain-dependent" from the sentence.

Reviewer #3

The MS by Mana et al is a careful study of the role of different $\beta 1$ integrin domains in integrin conformational changes and integrin clustering in fibrillar adhesion formation and cell migration. The authors use three different $\beta 1$ integrin mAbs, with well-defined epitopes, to demonstrate that the formation of fibrillar adhesions depends on $\beta 1$ domain and integrin clustering. The importance of FAK for integrin turnover in fibrillar adhesions is also demonstrated.

I have no comments as to experimental design or figures with excellent figure legends but have the following comments.

1. The authors in figure one show $\alpha 5\beta 1$ and mention its role in fibrillar adhesion formation. Throughout the study only $\beta 1$ integrin antibodies are used.

In Fig. 1A, we showed $\alpha 5\beta 1$ because it is the integrin that **mostly mediates the binding to fibronectin** (which we use as coating) in endothelial cells. In the revised version of the paper, however, we also verified what happens to the **active $\alpha 5$ integrin subunit**, employing the **SNAKA51 mAb as a reporter** (Clark et al, 2005, J Cell Sci 118: 291–300). Incubating **12G10 mAb on live endothelial cells also dismantles the localization in fibrillar adhesions of the SNAKA51⁺ active $\alpha 5$ integrin subunit**, followed by its internalization (**Page 6 and 11; Fig. 1E and Suppl. Figs. 2A and 4**).

2. The authors use EC throughout the MS to study fibrillar adhesion formation. In vivo endothelial cells sit on abasement membrane and under physiological conditions probably only to a limited degree are involved in fibrillar adhesion formation.

We agree with the Reviewer, yet, as shown by Richard Hynes and collaborators, **fibronectin** (George, et al., 1997, Blood 90:3073–3081) and **$\alpha 5\beta 1$ integrin** (Francis et al., Arterioscler. Thromb. Vasc. Biol., 2002, 22:927-933) are crucial for **developmental angiogenesis** in mouse embryos (Astrof & Hynes, 2009, Angiogenesis 12:165–175). Furthermore, fibronectin fibrillogenesis was found to regulate **3D blood vessel formation also in vitro** (Zhou, X. et al., 2008, Genes Dev. 22:1231–1243).

3. For future studies, using other cell types more traditionally involved in fibrillar adhesion formation, like fibroblasts, it would be useful to know what the set-up of $\beta 1$ integrins are in the cells studied. More specifically, what other fibronectin-binding $\beta 1$ integrins do these cells express?

Looking at the effect of 12G10 on fibrillar adhesion disassembly in cell types more traditionally involved in the formation of the same structures would be surely interesting. It is well established (Astrof and Hynes, 2009, Angiogenesis, 12:165-175) that the main fibronectin binding integrins in endothelial cells are **$\alpha 5\beta 1$ and $\alpha v\beta 3$ integrins, with a contribution of $\alpha v\beta 5$ as well.**

4. The authors use $\beta 1$ integrin antibodies to determine the domains in beta1 chain in the $\alpha 5\beta 1$ heterodimer which is involved in fibrillar adhesion formation. The authors should also perform staining with alpha5 antibody, like SNAKA51, to confirm staining pattern.

We also verified what happens to the **active $\alpha 5$ integrin subunit**, employing the **SNAKA51 mAb as a reporter** (Clark et al, 2005, J Cell Sci 118: 291–300). Incubating **12G10 mAb on live endothelial cells also dismantles the**

Localization in fibrillar adhesions of the SNAKA51⁺ active $\alpha 5$ integrin subunit, followed by its internalization (Page 6 and 11; Fig. 1E and Suppl. Figs. 2A and 4).

5. *In the discussion the authors should mention recent results in superresolution microscopy suggesting that fibrillar adhesions are composed of nanodomains (1). Is there a difference between $\alpha 5\beta 1$ integrin immunostaining in this regard?*

Reference cited: (1) Tomer, D. et al. New mechanism of fibronectin fibril assembly revealed by live imaging and super-resolution microscopy. *bioRxiv* <https://doi.org/10.1101/2020.09.09.290130> (2020).

We agree with the Reviewer, it would be interesting to precisely analyze the size of minimal clusters of active $\alpha 5\beta 1$ integrins and their organization in endothelial cells upon incubation with 12G10. However, since **the transverse diameter of an active $\alpha 5\beta 1$ integrin dimer (~ 12 nm) is comparable to the height of an intact antibody (~ 15 nm)**, we would need to generate and fluorescently label large amounts of anti-active integrin Fabs to be employed as reporters. This would allow us to avoid using fluorescently labeled secondary Abs and the troubles of nanoscopic measurements based on the use of intact Abs. We have planned to develop and set up these tools in the future.

6. *Immunostaining for $\beta 3$ integrin would be informative since it has been reported to localize at anchoring points of fibrillar adhesions; is this localization retained when fibrillar adhesion integrity is disturbed with $\beta 1$ integrin antibodies?*

We tested the effects of 12G10 on $\beta 3$ integrin. We observed incubation of 12G10 mAb on non confluent endothelial cells plated on vitronectin, which **does not affect number, mean area or mFD** of $\beta 3$ integrin⁺ adhesive sites (**Page 6 and Suppl. Fig. 2B**).

7. *Do treatment of cells with 12G10 also affect fibronectin assembly?*

Since clustered active $\alpha 5\beta 1$ integrins are tethering sites that promote the polymerization of soluble FN dimers into an insoluble fibrillar network (Schwarzbauer & DeSimone, 2011, *Cold Spring Harb Perspect Biol*, 3:a005041.), **we assessed the influence of 12G10 on the incorporation of soluble FN incorporation into polymeric FN fibrils.** To this aim, **rhodamine-labelled soluble FN** was added on cultured ECs for 1 hour and then followed or not by a 10 min incubation with 12G10, fixation and analysis by confocal fluorescence microscopy. We found that **12G10 clearly disrupted the incorporation of soluble rhodamine-FN into fibrils**, while promoting its accumulation into punctate structures (**Page 7 and Suppl. Fig. 2C**) that were bona fide endosomes.

October 28, 2022

RE: Life Science Alliance Manuscript #LSA-2022-01388-TR

Prof. Guido Serini
University of Turin
Department of Oncology
Candiolo Cancer Institute- FPO-IRCCS
Candiolo, Torino 10060
Italy

Dear Dr. Serini,

Thank you for submitting your revised manuscript entitled "The β 1 domain promotes active β 1 integrin clustering into mature adhesion sites". We would be happy to publish your paper in Life Science Alliance pending final revisions necessary to meet our formatting guidelines.

- please address Reviewer 2's comment and if the mentioned data are available, please include them as suggested
- please add a conflict of interest statement to your main manuscript
- In Data Availability section there is no need to refer to your source data

Figure Check:

- Figure S5 shouldn't be included as a Figure and should instead be uploaded as a source data file

A. FINAL FILES:

B. MANUSCRIPT ORGANIZATION AND FORMATTING:

Sincerely,

Reviewer #2 (Comments to the Authors (Required)):

Overall the authors have done a good job of addressing the points that I raised in my first review. As to the key point, does this work prove that it is the disruption of clustering that explains the inhibitory effects of some of the activating antibodies, they have significantly strengthened the evidence in favor of this model. The FAB experiment helps, but if that argument is correct than non activating antibodies should also suppress the effect of 12G10, since they will still dimerize the integrins. If they have this data they should add it. The demonstration that another antibody that binds to a similar place has the same effect is an important addition. Some independent verification of altered interactions/clustering between integrins would have been great, but I appreciate that its not straightforward to assay this.

Reviewer #3 (Comments to the Authors (Required)):

The MS by Mana et al is a careful study of the role of different $\beta 1$ integrin domains in integrin conformational changes and integrin clustering in fibrillar adhesion formation and cell migration. The authors use three different $\beta 1$ integrin mAbs, with well-defined epitopes, to demonstrate that the formation of fibrillar adhesions depends on $\beta 1$ domain and integrin clustering. The importance of FAK for integrin turnover in fibrillar adhesions is also demonstrated. I am happy with the rebuttal to my original comments and only have the following comments:

The authors use EC throughout the MS to study fibrillar adhesion formation. In vivo endothelial cells sit on a basement membrane and under physiological conditions probably only to a limited degree are involved in fibrillar adhesion formation.

We agree with the Reviewer, yet, as shown by Richard Hynes and collaborators, fibronectin (George, et al., 1997, Blood 90:3073-3081) and $\alpha 5\beta 1$ integrin (Francis et al., Arterioscler. Thromb. Vasc. Biol., 2002, 22:927-933) are crucial for developmental angiogenesis in mouse embryos (Astrof & Hynes, 2009, Angiogenesis 12:165-175).

This is correct and applies to very early steps of forming blood vessels, but early during embryogenesis as basement membranes form, $\alpha 6\beta 1$ integrin appears on endothelial cells and $\alpha 5\beta 1$ expression becomes more prominent on forming muscle and fibroblasts. With the growing realization of fibroblast and Cancer-associated fibroblasts(CAF) heterogeneity in fibrosis and cancer, it will in the future be interesting to determine if all fibroblast/CAF subsets expressing $\alpha 5\beta 1$ also contribute to FN fibrillogenesis.

Furthermore, fibronectin fibrillogenesis was found to regulate 3D blood vessel formation also in vitro (Zhou, X. et al., 2008, Genes Dev. 22:1231-1243).

This is not surprising since many integrins, including $\alpha 5 \beta 1$, can be induced during in vitro conditions, but in combination with in vivo data this does probably indicate a biologically relevant process.

Reviewer #2

Overall the authors have done a good job of addressing the points that I raised in my first review. As to the key point, does this work prove that it is the disruption of clustering that explains the inhibitory effects of some of the activating antibodies, they have significantly strengthened the evidence in favor of this model. The FAB experiment helps, but if that argument is correct than non activating antibodies should also suppress the effect of 12G10, since they will still dimerize the integrins. If they have this data they should add it. The demonstration that another antibody that binds to a similar place has the same effect is an important addition. Some independent verification of altered interactions/clustering between integrins would have been great, but I appreciate that its not straightforward to assay this.

We agree with the Reviewer that non-activating anti- β 1 integrin antibodies (Abs) might suppress the disrupting effect of 12G10 mAb. However, until now we have not yet tested this hypothesis and we do not have these data. We have planned to further test the ability of different anti- α 5 and anti- β 1 integrin subunit Abs to counteract the disrupting effects of 12G10 and TS2/16 mAbs in the future.

November 3, 2022

RE: Life Science Alliance Manuscript #LSA-2022-01388-TRR

Prof. Guido Serini
University of Turin
Department of Oncology
Strada Provinciale 142, Km 3.95
Candiolo, Torino 10060
Italy

Dear Dr. Serini,

Thank you for submitting your Research Article entitled "The β I domain promotes active β 1 integrin clustering into mature adhesion sites". It is a pleasure to let you know that your manuscript is now accepted for publication in Life Science Alliance. Congratulations on this interesting work.

DISTRIBUTION OF MATERIALS:

Again, congratulations on a very nice paper. I hope you found the review process to be constructive and are pleased with how the manuscript was handled editorially. We look forward to future exciting submissions from your lab.

Sincerely,
